# CXCL5 activates CXCR2 in nociceptive sensory neurons to drive joint pain and inflammation in experimental gouty arthritis

Chengyu Yin [1,9], Boyu Liu [1,9], Zishan Dong[2,9], Sai Shi[3,9], Chenxing Peng[4], Yushuang Pan[1], Xiaochen Bi[5], Huimin Nie[1], Yunwen Zhang[1], Yan Tai[6], Qimiao Hu[1], Xuan Wang[7], Xiaomei Shao[1], Hailong An [8] ✉, Jianqiao Fang [1] ✉, Chuan Wang [2] ✉ & Boyi Liu [1] ✉

Gouty arthritis evokes joint pain and inflammation. Mechanisms driving gout pain and inflammation remain incompletely understood. Here we show that CXCL5 activates CXCR2 expressed on nociceptive sensory neurons to drive gout pain and inflammation. CXCL5 expression was increased in ankle joints of gout arthritis model mice, whereas CXCR2 showed expression in joint-innervating sensory neurons. CXCL5 activates CXCR2 expressed on nociceptive sensory neurons to trigger TRPA1 activation, resulting in hyperexcitability and pain. Neuronal CXCR2 coordinates with neutrophilic CXCR2 to contribute to CXCL5-induced neutrophil chemotaxis via triggering CGRP- and substance P-mediated vasodilation and plasma extravasation. Neuronal Cxcr2 deletion ameliorates joint pain, neutrophil infiltration and gait impairment in model mice. We confirmed CXCR2 expression in human dorsal root ganglion neurons and CXCL5 level upregulation in serum from male patients with gouty arthritis. Our study demonstrates CXCL5-neuronal CXCR2-TRPA1 axis contributes to gouty arthritis pain, neutrophil influx and inflammation that expands our knowledge of immunomodulation capability of nociceptive sensory neurons.

Gout arthritis is the most common inflammatory arthritis worldwide[1,2]. It is triggered by monosodium urate crystals (MSU) deposition in the joints[1]. Patients with gout arthritis develop joint swelling and suffer from intense pain that dramatically affects daily activity and life quality[3,4]. Unfortunately, the incidence of gout arthritis is constantly increasing due to the aging population, obesity, and dietary changes. For the management of acute gout attacks, patients are prescribed with colchicine, nonsteroidal anti-inflammatory drugs, or corticosteroids[1]. However, constant use of these drugs results in severe side effects, especially among patients with comorbidities like chronic

[1]Key Laboratory of Acupuncture and Neurology of Zhejiang Province, Department of Neurobiology and Acupuncture Research, the Third Clinical Medical College, Zhejiang Chinese Medical University, Hangzhou, China. [2]Department of Pharmacology, Hebei Medical University, Shijiazhuang, China. [3]Frontiers Science Center for Synthetic Biology (Ministry of Education), Tianjin Key Laboratory of Function and Application of Biological Macromolecular Structures, School of Life Sciences, Tianjin University, Tianjin, China. [4]Department of Immunology and Rheumatology, the Second Hospital of Hebei Medical University, Shijiazhuang, China. [5]Department of Human Anatomy, School of Basic Medical Sciences, Zhejiang Chinese Medical University, Hangzhou, China. [6]Academy of Chinese Medical Sciences, Zhejiang Chinese Medical University, Hangzhou, China. [7]Diagnostic Center of Infections, The Second Hospital of Hebei Medical University, Shijiazhuang, China. [8]Key Laboratory of Molecular Biophysics, Hebei Province, Institute of Biophysics, School of Sciences, Hebei University of Technology, Tianjin, China. [9]These authors contributed equally: Chengyu Yin, Boyu Liu, Zishan Dong, Sai Shi. ✉e-mail: hailong_an@hebut.edu.cn; fangjianqiao7532@163.com; wangchuan@hebmu.edu.cn; boyi.liu@foxmail.com

kidney disease or diabetes, which limits their frequent usage[3,5,6]. IL-1β neutralizing antibodies or IL-1R antagonists are new options, yet patients still need to be watched closely for intolerabilities and side effects[7,8]. Therefore, identifying a novel therapeutic target for gout arthritis is of huge clinical significance.

MSU deposition in the joint triggers immune activation. Tissue-resident macrophages engulf MSU via phagocytosis and release a panel of cytokines or chemokines (e.g., IL-1β, TNF-α)[9–11]. These cytokines or chemokines drive neutrophil recruitment in the joint, which is a hallmark of acute gout arthritis[12,13]. The influx of neutrophils into joints can release ROS and inflammatory mediators that further contribute to pain and tissue damage[12,14,15]. However, the mechanism underlying how pain signal is generated in inflamed joints in the context of gout arthritis still remains incompletely understood.

Chemokine CXCL5 belongs to the CXC chemokine family. It is an inflammatory mediator and a powerful neutrophil chemoattractant[16]. It mainly acts through CXCR2 to produce biological effects, although it also has weak activity on CXCR1[17,18]. CXCL5 is demonstrated to contribute to sunburn pain by recruiting inflammatory cell infiltrations[19]. CXCR2 is abundantly expressed on neutrophils and is important for neutrophil chemotaxis[20]. In addition to the immune system, CXCR2 expression is also found in peripheral sensory neurons and spinal cord, which may contribute to pain mechanism[21,22].

Here, we identified that CXCL5 was among the top chemokines overexpressed in ankle joints of a mouse gout arthritis model. It acts upon CXCR2 in nociceptor neurons to trigger TRPA1 activation, resulting in membrane depolarization and hyperexcitability. By employing sensory neuron-specific deletion of *Cxcr2*, we found neuronal CXCR2 mediates CXCL5-induced pain and neurogenic inflammation. Neuronal CXCR2 coordinates with neutrophil CXCR2 to contribute to CXCL5-induced neutrophil chemotaxis by promoting calcitonin gene-related peptide (CGRP) and substance P (SP) release and plasma extravasation. Neuronal-specific *Cxcr2* deletion ameliorates joint pain, neutrophil infiltration, and gait impairments in model mice. CXCR2 is expressed in human dorsal root ganglion (DRG) neurons and CXCL5 is increased in serum from gout arthritis patients and correlated positively with serum urate level. Our findings demonstrate that CXCL5 activates CXCR2 in nociceptor neurons to drive acute gout arthritis pain and joint inflammation. Targeting CXCR2 in joint-innervating nociceptor neurons could be a potential therapy for gout arthritis.

## Results

### CXCL5 is elevated in inflamed ankle joints of gout arthritis model mice and contributes to joint pain and inflammation
We established a mouse model of gout arthritis by injecting MSU into the ankle joint as described[23,24]. Control mice received PBS injections. Model mice showed an obvious increase in ankle diameter and mechanical allodynia vs. control mice (Fig. 1A–C). To search signaling pathways that may contribute to gout arthritis pain and joint inflammation, we performed gene expression profiling in ankle joints via RNA sequencing (RNA-Seq). Samples were collected at 8 and 24 h time points when model mice developed severe pain and inflammation. RNA-Seq identified a number of differentially expressed genes (DEGs) at these two-time points (Supplementary Fig. 1). We crossed these two datasets to obtain a core set of DEGs that were simultaneously altered from 8 to 24 h time points (Fig. 1D).

Bioinformatics analysis of this core set of DEGs identified several enriched signaling pathways, among which cytokine–cytokine receptor interaction ranked the top (Fig. 1E). Cytokine–cytokine receptor interaction plays an important role in mediating pain and inflammation[25]. We checked the full list of DEGs in this pathway (Fig. 1F). Of particular interest was chemokine CXCL5, which was among the top upregulated genes. CXCL5 has not been previously implicated in gout pain but has been shown to participate in sunburn

pain[19]. qPCR confirmed *Cxcl5* gene upregulation in ankle joints of model mice (Fig. 1G). ELISA showed CXCL5 expression was increased in ankle joints of model mice 8, 24, and 48 h after model establishment (Fig. 1H). We then explored the cellular source of CXCL5. CXCL5 can be produced by macrophages, mast cells, fibroblasts, etc.[26–28]. and these cells can be activated by MSU[14,29,30]. Immunostaining of ankle joint from gout model mice showed that a majority (56.6%) of CXCL5+ cells co-stained with fibroblast cell marker vimentin and to a lesser extent, co-stained with markers for macrophage (Iba-1) and mast cell (avidin) (30.0% and 12.3%, respectively) (Supplementary Fig. 2). This result indicates CXCL5 can be produced from several types of cells in the inflamed joint, including fibroblasts, macrophages and mast cells, during gout arthritis.

Neutralizing antibodies against CXCL5, CXCL2, or CXCL1 or isotype control IgG were injected (3 μg/site) into the ankle joint. Neutralizing CXCL5 remarkably attenuated mechanical allodynia and ankle diameter of model mice vs. control IgG (Fig. 1I and J). However, neutralizing CXCL1 or CXCL2 only showed ameliorating effects on ankle joint diameter but not on pain (Fig. 1I and J). CXCL5 neutralizing antibody significantly reduced neutrophil and macrophage infiltrations in ankle joints of gout model mice, indicated by immunostaining and myeloperoxidase (MPO) assay (Supplementary Fig. 3A–E). These results suggest chemokine CXCL5 contributes to pain and joint inflammation in gout model mice.

### CXCL5 receptor CXCR2 is expressed in mouse ankle joint-innervating DRG neurons
We hypothesize that increased CXCL5 in ankle joints of gout model mice might act upon its receptors expressed on peripheral sensory neurons to produce pain. We checked the expression of *Cxcr1* and *Cxcr2* genes in mouse DRG. PCR from mouse DRG only detects *Cxcr2* but not *Cxcr1* expression. As a positive control, both genes were detected in the mouse spleen (Fig. 2A). This result rules out *Cxcr1* expression in DRG neurons. Previous studies documented CXCR2 expression in DRG neurons[22,31]. We further examined whether CXCR2 was expressed in ankle joint-innervating DRG neurons labeled by prior injection of retrograde tracer WGA into ankle joint[32]. WGA-labeled neurons (WGA+) take up 28.9% of total DRG neurons (NeuN+). CXCR2 was expressed in 92.7% of NeuN+ neurons and in 97.3% of WGA-labeled ankle-joint innervating DRG neurons (Fig. 2B and C). CXCR2 was exclusively co-expressed with neuronal marker NeuN but barely with satellite glia cell marker GFAP (Fig. 2B, C, and E). The specificity of the CXCR2 antibody was confirmed by the absence of immunostaining in DRG from *Cxcr2−/−* mice (Fig. 2D). We mapped the CXCR2 expression profile in ankle joint-innervating DRG neurons with this specific antibody and co-stained with markers for different cell populations. Figure 2F showed CGRP, IB4, and NF200 were expressed in 51.6%, 30.1%, and 27.6% of CXCR2+WGA+ cells, respectively. These data demonstrated that ankle joint-innervating sensory neurons, including nociceptive sensory neurons, have CXCR2 expression, providing a possible molecular basis for CXCL5's interaction with neuronal CXCR2 in the context of gout arthritis.

### CXCL5/CXCR2 signaling functionally couples to TRPA1 channel in mouse DRG neurons
Exposure of cultured DRG neurons to recombinant mouse CXCL5 (rmCXCL5, 100 nM) triggered obvious Ca2+ transients in a subgroup of DRG neurons, which accounts for 26.0% among all neuron populations (Fig. 3A–C and F). 79.3% of CXCL5-responding (CXCL5+) neurons showed responsiveness to allyl isothiocyanate (AITC, 100 μM, a TRPA1 agonist), whereas 69.7% of CXCL5+ neurons showed responsiveness to capsaicin (TRPV1 agonist, 300 nM) application (Fig. 3B and C). The magnitude of Ca2+ response induced by CXCL5 was inversely correlated with neuron size; namely, stronger response was usually recorded from small to medium-sized neurons (Fig. 3D). These data suggest

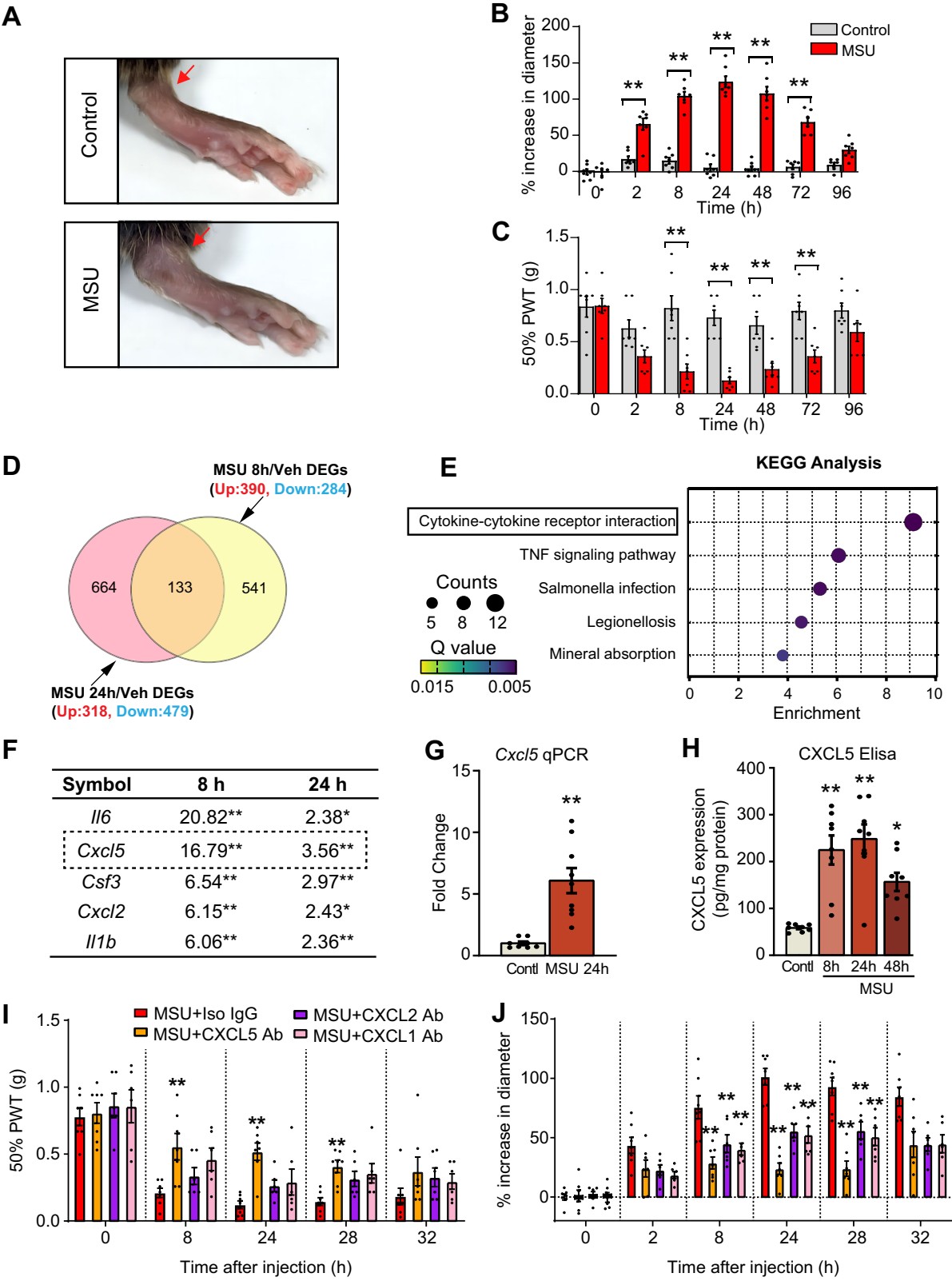

CXCL5-responding neurons were predominantly nociceptive sensory neurons. CXCL5 dose-dependently triggered Ca²⁺ response in mouse DRG neurons (EC$_{50}$ = 24.8 ± 1.9 nM) (Fig. 3E). We next compared the effects of CXCL5 with other chemokines that similarly target CXCR2, including CXCL1 and CXCL2. When tested at the same dosage (100 nM), CXCL1 and CXCL5 both produced Ca²⁺ response in mouse

DRG neurons. CXCL5's response is significantly higher than CXCL1, whereas CXCL2 has no effect (Fig. 3F).

CXCL5's effect was significantly reduced when it was denatured by boiling or by pretreatment with CXCR2 specific antagonist SB225002 (100 nM) or in neurons derived from *Cxcr2*⁻/⁻ mice (Fig. 3G, left panel), suggesting CXCR2 was the prime receptor for CXCL5 in DRG neurons.

**Fig. 1 | CXCL5 is a highly upregulated chemokine in the inflamed ankle joint of gout arthritis model mice and contributes to joint pain and inflammation.**
**A** Representative picture showing the ankle joint injected with PBS (control group) or MSU (MSU group). The pictures were taken 24 h after injection by using the camera. **B** Delta changes of ankle diameter of control and MSU group of mice in time course. **C** 50% PWT changes of control and MSU group of mice in time course. **D** Venn diagram illustrating the overlapped DEGs identified from 8 and 24 h time points. The dataset of 24 h time points was derived from our recent study[14] and reanalyzed accordingly. **E** KEGG analysis of overlapped DEGs identified in panel (**E**). **F** List of top ranking genes included in cytokine–cytokine receptor interaction as in panel (**E**). The absolute fold changes of these genes are shown. **G** qPCR test of *Cxcl5*

gene expression in control and MSU 24 h group. **H** CXCL5 ELISA test of ankle joint tissue homogenates from control and 8, 24, 48 h after MSU injection **$p < 0.01$, *$p < 0.05$ vs. control. **I** Effect of CXCL1, CXCL2, or CXCL5 neutralizing antibody (3 μg/5 μl) or isotype control IgG on mechanical allodynia of model mice. **J** Effect of neutralizing antibodies or isotype control IgG on ankle diameter of model mice. **$p < 0.01$, *$p < 0.05$ vs. MSU + Iso IgG group. Two-way ANOVA (repeated measures) with Bonferroni's post hoc test was used (panels **B**, **C**, **I**, and **J**). One-way ANOVA with Bonferroni's post hoc test was used for panel (**H**). Student's unpaired *t*-test (two-tailed) was used for panel (**G**). Data are shown as mean ± SEM. The *n* number, exact *p*-value, and statistical results are provided in the Source Data file.

Similar results were also obtained from CXCL1-challenged neurons (Fig. 3G, right panel). Since CXCL5 produces a significantly higher Ca²⁺ response than CXCL1 and makes significant contributes to gout arthritis pain, we therefore especially focused on CXCL5 in our following studies. Replacing extracellular solution (ES) with Ca²⁺-free ES (Fig. 3H) or treating cells with broad spectrum TRP channel blocker ruthenium red (RR, 10 μM, Fig. 3I) largely abolished capsaicin-, AITC- and CXCL5-induced Ca²⁺ responses, suggesting TRP channels as possible downstream targets of CXCL5/CXCR2 signaling in mouse DRG neurons. We reasoned that CXCL5/CXCR2 may couple to TRPV1 or TRPA1. We then tested this hypothesis by treating cells with AMG9810 (1 μM) or HC030031 (100 μM) to specifically block TRPV1 or TRPA1. CXCL5-induced Ca²⁺ response was significantly reduced by HC030031 but not by AMG9810 (Fig. 3J). We next tested CXCL5's effect on DRG neurons derived from *Trpv1*⁻/⁻ and *Trpa1*⁻/⁻ mice. CXCL5-induced Ca²⁺ response was almost completely abolished in neurons derived from *Trpa1*⁻/⁻ but not from *Trpv1*⁻/⁻ mice (Fig. 3K and L). Capsaicin- or AITC-induced Ca²⁺ response was abolished in *Trpv1*⁻/⁻ and *Trpa1*⁻/⁻ mice, respectively, demonstrating the validity of the knockout mouse strains (Fig. 3K and L). We explored whether CXCR2 showed spatial co-localization with TRPA1. RNAscope combined with immuno-fluorescence showed *Trpa1* gene was expressed in 44.2% of CXCR2 positively stained (CXCR2⁺) neurons in mouse DRG, providing strong anatomical support for CXCR2 and TRPA1 coupling in nociceptive sensory neurons (Fig. 3M).

**Exploring the biochemical basis underlying CXCR2-mediated TRPA1 activation**
We then asked whether CXCL5-CXCR2-TRPA1 signaling identified from mouse DRG neurons could be reconstituted in a heterologous expression system. Application of rmCXCL5 (100 nM) failed to evoke any Ca²⁺ response in HEK293T cells transfected with plasmids containing empty vector, m*Cxcr2* (mouse *Cxcr2*) or m*Trpa1* alone or m*Cxcr1*+m*Trpa1* (Fig. 4A–F). In contrast, HEK293T cells transfected with m*Cxcr2*+m*Trpa1* showed robust Ca²⁺ responses upon CXCL5 application (Fig. 4A–F, E and F), suggesting both CXCR2 and TRPA1 are needed to confer CXCL5's effect on HEK293T cells.

Previous work suggests Gβγ could serve as a downstream signaling of GPCR to activate TRPA1[33,34]. But how exactly Gβγ works to activate TRPA1 remains unknown. We made use of the expression system to test whether sequestration of endogenous Gβγ subunits can block mTRPA1 activation by mCXCR2. Transfecting HEK293T cells with either Gαtransducin (Gαt) or the carboxy-terminal domain of G protein-coupled receptor kinase 2 (GRK2ct), which are both sca-vengers of Gβγ subunits[35–37], significantly reduced mTRPA1 activation by mCXCR2 upon CXCL5 challenge (Fig. 4G). We further used siRNA to knockdown the most abundantly expressed Gβ isoforms, Gβ1 and Gβ2, in HEK293T cells[38]. *Gnb1* and *Gnb2* gene knockdown significantly reduced mTRPA1 activation by mCXCR2 upon CXCL5 challenge (Fig. 4G). AITC activates TRPA1 via direct covalent modification of cysteine residuals in TRPA1 channel[39]. We found that neither seques-tration of endogenous Gβγ nor knockdown of *Gnb1* and *Gnb2* expression has any significant effect on AITC-induced TRPA1 activation

(Fig. 4H). These results indicate that CXCR2 triggers TRPA1 activation via Gβγ signaling.

After the dissociation from Gα upon activation, Gβγ can modulate ion channel activity via direct binding to the channel or via triggering downstream signaling[40,41]. We tested if Gβγ-related downstream sig-naling contributes to TRPA1 activation by CXCR2. Pharmacological blocking Gβγ-related downstream signaling, including phospholipase C (PLC), phosphoinositide 3-kinase (PI3K), protein kinase A (PKA), or mitogen-activated protein kinase (MAPK) at effective dosages has no significant effect on mTRPA1 activation by mCXCR2 upon CXCL5 challenge. Similarly, blocking these pathways has no effect on AITC-induced TRPA1 activation (Fig. 4H). To further explore if Gβγ directly binds with TRPA1 upon CXCR2 activation, we performed co-immunoprecipitation (Co-IP) assay. Co-IP using anti-TRPA1 antibody revealed stronger coupling of Gβ with TRPA1 upon CXCR2 activation by CXCL5. (Fig. 4I and J). These results indicate Gβγ can bind with TRPA1 upon CXCR2 activation.

To further explore the structure basis of CXCR2 regulation of TRPA1, we performed structure-based modeling to predict mTRPA1 and Gβγ binding. The docking model shows that Gβ binds to the intracellular structural domain of mTRPA1, with K992-M1018 in mTRPA1 being the region of direct interaction, and the secondary structure of this region shows a helix and β-sheet (Fig. 4K–N). The region we predicted where mTRPA1 binds to Gβ is located at the C-terminal end of the TRP helix, and Gβ binds to TRPA1 in a spatially similar location as it does to TRPM3, both being located on the lower side of the transmembrane structural domain and at the largest part of the protein radius as recently reported[42]. Electrostatic interaction analysis showed that the K992-M1018 segment of mTRPA1 exhibits a high potential feature, which perfectly complements the low potential of the Gβ-interacting region (Fig. 4N, regions A and B). As shown in Fig. 4O, we identified five key residues involved in the electrostatic interaction of mTRPA1 with Gβ, including R1012/R1017 in region A and R999/K1000/R1004 in region B. Similar to the working model for the TRPM3–Gβγ complex, K992-M1018 of TRPA1 does not interact with spatially distant Gγ[42].

We then performed point mutations of these key residuals to see if they affect Gβ binding with mTRPA1 and subsequently affect mTRPA1 activation by mCXCR2. Mutation of the two residues (R1012 and R1017) located in region A only slightly reduced TRPA1 activation. Mutation of the three residues (R999, K1000, and R1004) located in region B significantly reduced mTRPA1 activation induced by mCXCR2. Furthermore, mutations of all five residues caused the most obvious reduction of mTRPA1 activation by mCXCR2 (Fig. 4P). We ascertained beforehand that all channel proteins carrying these point mutations were still functional since AITC evoked similar responses in wildtype and all mutated channels (Fig. 4P). Finally, we tested if the mutated TRPA1 channel exhibits reduced binding with Gβ by co-IP. The binding of TRPA1 with Gβ was markedly reduced by mutations of all 5 residues (Fig. 4Q). Based upon these results, we speculate that Gβγ binding to the K992-M1018 segment of TRPA1 causes outward move-ment of the TRP helix, which pulls TM6 to undergo metastasis and opens the lower gate of TRPA1 channel (Fig. 4R). Therefore, we identified the key regions and the residues located in TRPA1 that

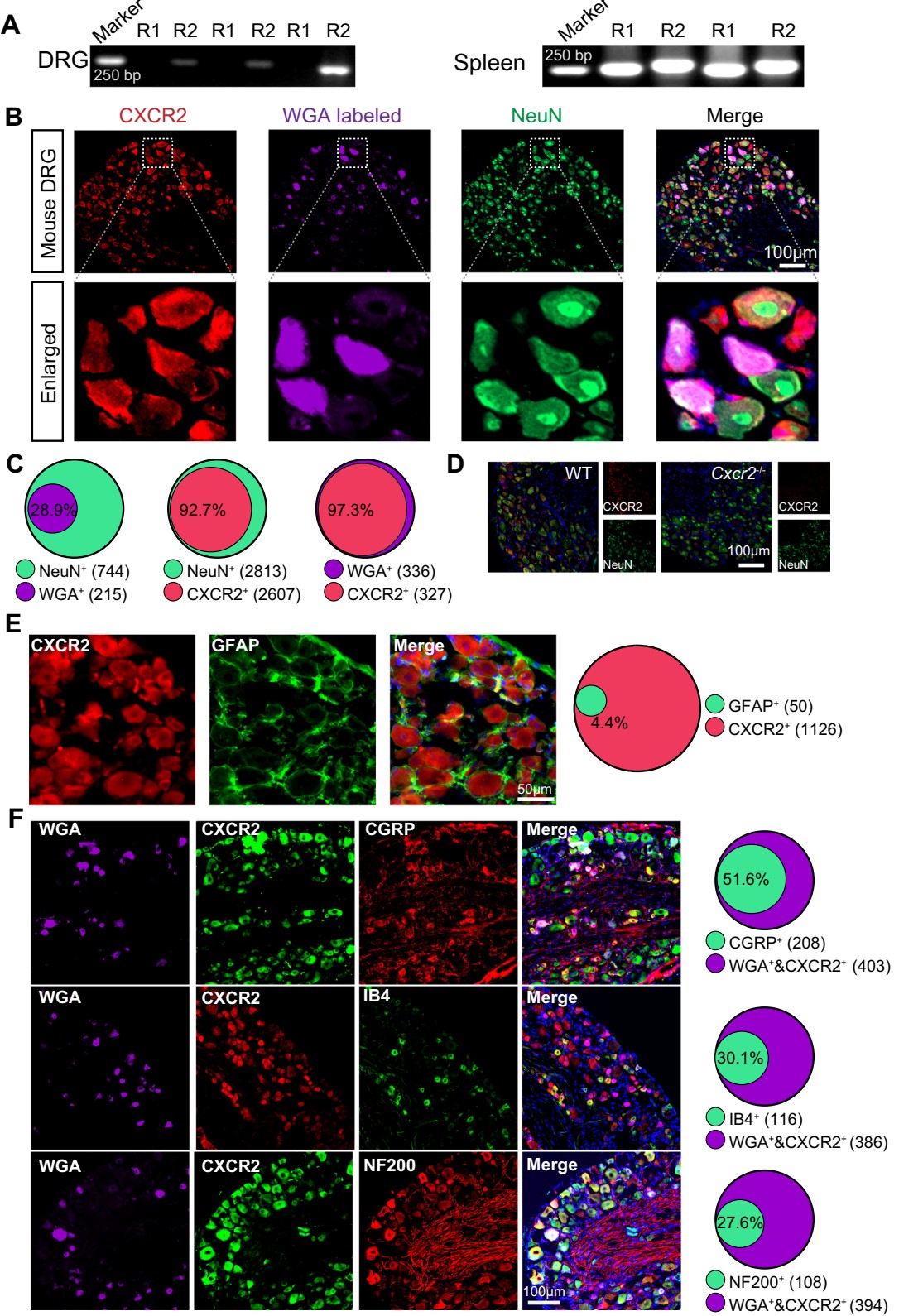

**Fig. 2 | CXCR2 is expressed in mouse sensory neurons innervating the ankle joint. A** PCR showing the expression of *Cxcr1* and *Cxcr2* genes in mouse DRG (left panel) and spleen (right panel). **B** Immunostaining showing CXCR2 expression in ankle joint-innervating DRG neurons labeled by retrograde tracer WGA. Scale bar = 100 μm. **C** Pie graphs showing the % of NeuN+ cells showing WGA+ (left), % of NeuN+ cells showing CXCR2+ (middle), and % of WGA+ cells showing CXCR2+ (right). The numbers in parenthesis indicate the number of cells included.
**D** Immunostaining showing the absence of CXCR2 staining in DRG neurons from

global *Cxcr2* gene-deficient (*Cxcr2-/-*) mice. Scale bar = 100 μm. **E** Immunostaining of CXCR2 with markers for satellite glial cells (GFAP) in mouse DRG. Right panel denotes quantification. Scale bar = 50 μm. **F** Immunostaining of CXCR2 with markers for non-peptidergic (IB4), peptidergic (CGRP), and large-diameter (NF200) sensory neurons in mouse DRG. Scale bar = 100 μm. Pie graph quantification is shown on the right. Three sections were obtained from each mouse, calculated, and averaged. Data were pooled from four mice/group.

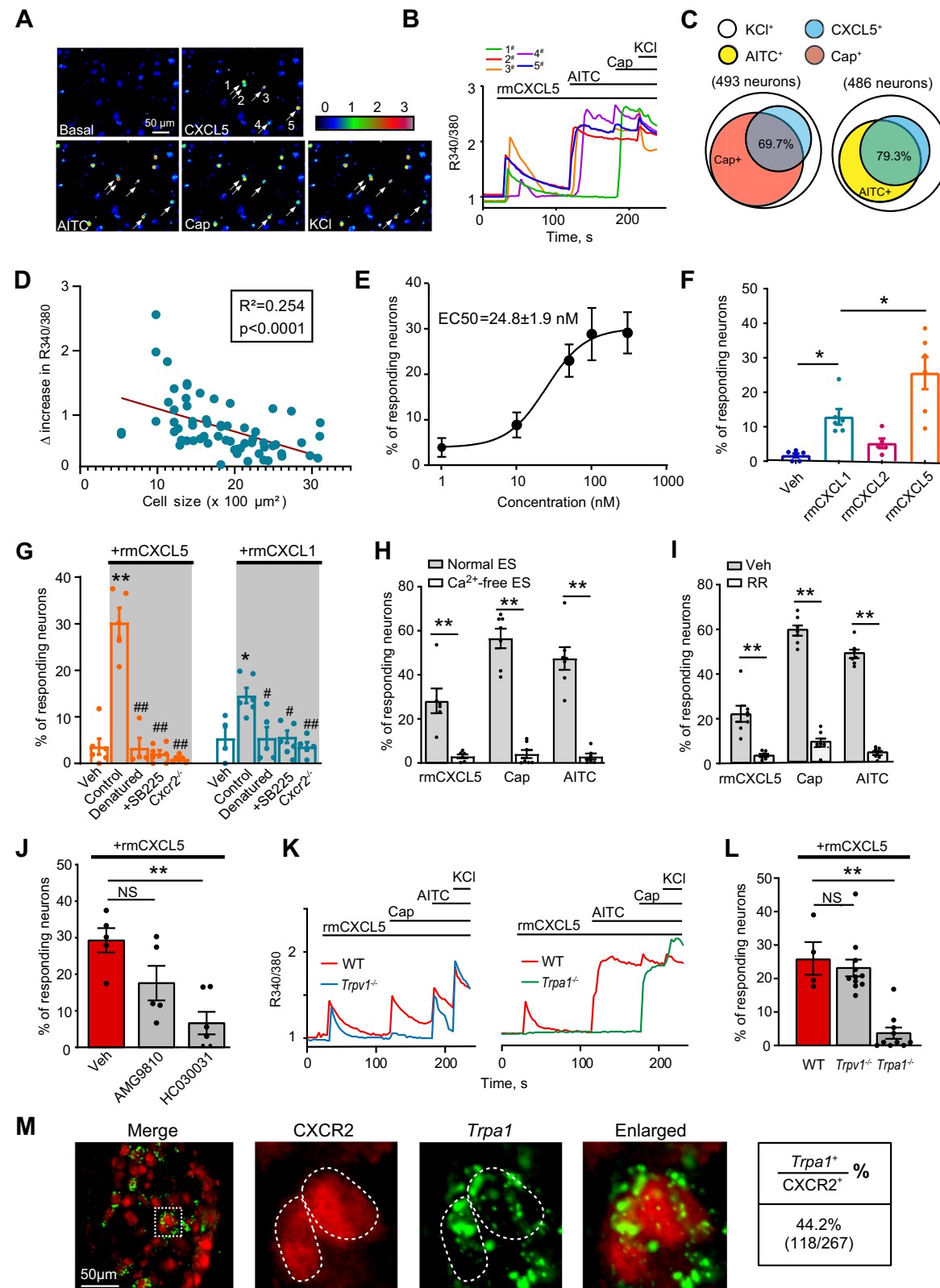

are involved in binding with Gβγ, leading to subsequent channel activation.

## CXCL5 depolarizes and excites nociceptive DRG neurons via CXCR2 and TRPA1-dependent mechanism

We tested the effects of CXCL5 on the excitability of mouse DRG neurons. We selected small-sized DRG (Cm < 42 pF) for recording as in

our recent study since a majority of these neurons are nociceptive[43,44]. Bath application of rmCXCL5 (100 nM) significantly increased action potential (AP) firing triggered by injecting a step of currents in DRG neurons vs. vehicle-treated neurons (Fig. 5A and B). Figure 5C−F shows the APs evoked by 150 pA current injection when a neuron is treated with vehicle or CXCL5. Some parameters of electrophysiology are listed in Supplementary Table 2. CXCL5-induced increases in AP firing

**Fig. 3 | CXCL5 acts upon CXCR2 to trigger TRPA1 activation in mouse DRG neurons. A** Ca²⁺ imaging showing Ca²⁺ responses of mouse DRG neurons upon CXCL5 (100 nM), AITC (100 μM), capsaicin (300 nM) and KCl (40 mM) challenges. Arrows indicate cells responding to CXCL5. Scale bar, 50 μm. **B** Ca²⁺ traces upon CXCL5 challenges recorded in panel **A**, with different colors denote corresponding neurons indicated by the numbers in panel (**A**). **C** Venn diagram illustrating the overlapping of CXCL5⁺ neurons with capsaicin⁺ (Cap⁺) or AITC⁺ neuron population. **D** The correlation of cell size with delta increase in R340/380 induced by CXCL5. **E** Dose–response curve of CXCL5 induced Ca²⁺ response. **F** % of DRG neurons responding to the vehicle (0.1% BSA) or 100 nM CXCL1, CXCL2, or CXCL5. Each recording field includes 40–60 neurons. **G** % of DRG neurons responding to the vehicle and CXCL5 (100 nM) or CXCL1 (100 nM) in control, boiled-treated, SB225002 (100 nM)-treated, and *Cxcr2⁻/⁻* conditions. **H** % of DRG neurons responding to CXCL5, capsaicin, and AITC recorded in normal extracellular solution (ES) or EGTA-buffered Ca²⁺-free extracellular solution (Ca²⁺-free ES). **I** % of DRG neurons responding to CXCL5, capsaicin, and AITC in Veh- or ruthenium red (RR, 10 μM)-treated condition. **J** % of DRG neurons responding to CXCL5 under the vehicle, AMG9810 (1 μM) or HC030031 (100 μM)-treated condition. **K** Representative Ca²⁺ traces of DRG neurons from WT, *Trpv1⁻/⁻* and *Trpa1⁻/⁻* mice upon CXCL5 challenge. **L** % of DRG neurons responding to CXCL5. **M** Representative images of *Trpa1* mRNA with CXCR2 in mouse DRG. Summary of co-expression is shown on the right. 267 neurons summarized from 6 observational from 3 mice were included. Scale bar, 50 μm. *$p < 0.05$, **$p < 0.01$, ##$p < 0.01$. NS: no significance. Linear regression analysis for panel (**D**). Nonlinear regression with curve fit (four parameters) for panel (**E**). One-way ANOVA with Bonferroni's post hoc test was used for panels (**F**, **G**, **J**, and **L**). Student's unpaired *t*-test (two-tailed) was used for panels (**H** and **I**). Data are shown as mean ± SEM. The n number, exact *p*-value, and statistical results are provided in the Source Data file.

were significantly reduced by bath application of CXCR2-specific antagonist SB225002 (100 nM) or TRPA1-specific antagonist HC030031 (100 μM) (Fig. 5G–J). CXCL5-induced increases in AP firing were also reduced in DRG neurons derived from *Cxcr2⁻/⁻* or *Trpa1⁻/⁻* mice (Fig. 5K–N). CXCL5 triggered obvious depolarization of the resting membrane potential in small-sized DRG neurons (Fig. 5O). Pharmacological blocking CXCR2 or TRPA1 or genetic deleting *Cxcr2* or *Trpa1* all significantly attenuated CXCL5-triggered membrane potential depolarization (Fig. 5O). Thus, CXCL5 depolarizes and excites mouse DRG neurons via CXCR2 and TRPA1-dependent mechanisms.

## CXCR2 expressed in nociceptive DRG neurons contributes to CXCL5-induced mechanical pain and joint inflammation

CXCL5 per se was reported to induce mechanical pain hypersensitivity and inflammatory cell infiltrations in naïve rats[19]. But its downstream signaling remains largely unknown. We tested whether CXCR2 was responsible for these effects by CXCL5. Intraarticular injection of CXCL5 (3–300 ng/ankle) produced mechanical allodynia and an increase in joint diameter in a dose-dependent manner in mice (Supplementary Fig. 4A and B). Since 300 ng CXCL5 produced the most obvious effect, we chose this dosage for subsequent studies. CXCL5-induced mechanical allodynia was nearly completely attenuated by pharmacological blockage of CXCR2 (Supplementary Fig. 4C). Furthermore, *Cxcr2⁻/⁻* mice exhibited significantly reduced mechanical allodynia and joint diameter vs. WT mice upon CXCL5 injection (Supplementary Fig. 4D and E).

To gain further insight into neuronal CXCR2's contribution to CXCL5-induced mechanical pain and inflammation, we generated a new mouse line with conditional deletion of *Cxcr2* in peripheral nociceptive sensory neurons expressing Nav1.8 using SNS-Cre[45,46] (Fig. 6A). Immunostaining confirmed the reduction of CXCR2 specifically in DRG neurons but not in spleen, peripheral blood neutrophils, or spinal cord of SNS-Cre *Cxcr2ᶠˡ/ᶠˡ* mice (Fig. 6B–D, Supplementary Fig. 5A–D). SNS-Cre *Cxcr2ᶠˡ/ᶠˡ* mice showed the normal percentage of different neuronal populations in DRG and normal peripheral and central innervations (Supplementary Fig. 6A–H) and did not show abnormal signs in weight, basal pain threshold to mechanical or heat stimulation and locomotor activity (Supplementary Fig. 6I–L). These results indicate conditional knockout of *Cxcr2* in nociceptive sensory neurons is successful.

We then examined whether CXCR2 in nociceptive neurons contributes to CXCL5-induced mechanical allodynia and joint inflammation. CXCL5-induced mechanical allodynia, joint inflammation, and increases in pro-inflammatory gene expression (e.g., *Il1b*, *Tnfa*, and *Il6*) were markedly attenuated in SNS-Cre *Cxcr2ᶠˡ/ᶠˡ* mice vs. SNS-Cre negative controls (*Cxcr2ᶠˡ/ᶠˡ*) (Fig. 6F–K). We continued to test whether TRPA1 was essential for CXCL5-induced effects. Compared with WT controls, CXCL5-induced mechanical allodynia was significantly improved in *Trpa1⁻/⁻* but not in *Trpv1⁻/⁻* mice (Supplementary Fig. 7A). CXCL5-induced joint inflammation and neutrophil infiltration was significantly attenuated in *Trpa1⁻/⁻* mice (Supplementary Fig. 7B and C). These results demonstrate neuronal CXCR2-TRPA1 signaling in nociceptive sensory neurons contributes to CXCL5-induced mechanical pain and joint inflammation.

## Neuronal CXCR2 contributes to CXCL5-induced neurogenic inflammation and neutrophil chemotaxis

Since CXCL5 acts upon neuronal CXCR2 to activate TRPA1, an ion channel involved in mediating neurogenic inflammation, we then examined whether CXCL5 might trigger neurogenic inflammation. Intraplantar CXCL5 (300 ng/site) elicited obvious neurogenic inflammation in *Cxcr2ᶠˡ/ᶠˡ* mice 3 h after injection, as shown by a significant increase in Evans blue (EB) extravasation in injected hind paws. CXCL5-induced EB extravasation was significantly reduced in SNS-Cre *Cxcr2ᶠˡ/ᶠˡ* mice or *Trpa1⁻/⁻* mice or in mice treated with CGRP receptor antagonists BIBN (60 ng/site, i.pl.) or SP NK-1 receptor antagonist L-733060 (1 μg/site, i.pl.) (Fig. 7A–C). Since we found CXCR2 largely co-expressed with CGRP⁺ DRG neurons by immunostaining (Fig. 2F), we tested whether CXCL5 could trigger CGRP and SP release, two neuropeptides involved in vasodilation and plasma extravasation[47], by activating neuronal CXCR2. Intraplantar CXCL5 (300 ng/site) resulted in an obvious increase in CGRP and SP levels in the hind paws of *Cxcr2ᶠˡ/ᶠˡ* mice 3 h after injection vs. vehicle-treated mice (Fig. 7D and E), whereas these increases were significantly reduced in SNS-Cre *Cxcr2ᶠˡ/ᶠˡ* mice (Fig. 7D and E).

CXCL5 exerts strong chemotaxis on neutrophils by acting on neutrophil CXCR2[48]. Since we found that CXCL5 acts upon neuronal CXCR2 to promote plasma extravasation, it is tempting to know whether neuronal CXCR2 may contribute to CXCL5's neutrophil chemotaxis by promoting plasma extravasation. Intraplantar CXCL5 injection promoted significant neutrophil recruitment in injected hind paws of *Cxcr2ᶠˡ/ᶠˡ* mice, as revealed by both the MPO test and immunostaining (Fig. 7F, Supplementary Fig. 7E and F). CXCL5-induced neutrophil recruitment was significantly reduced in SNS-Cre *Cxcr2ᶠˡ/ᶠˡ* (Fig. 7F, Supplementary Fig. 7E, F). CXCL5 injection into the hind paw did not significantly affect serum CXCL5 level in either *Cxcr2ᶠˡ/ᶠˡ* or SNS-Cre *Cxcr2ᶠˡ/ᶠˡ* mice, ruling out the possibility that SNS-Cre *Cxcr2ᶠˡ/ᶠˡ* may alter blood-tissue CXCL5 gradient after CXCL5 injection (Supplementary Fig. 7G). These results suggest an important contribution of neuronal CXCR2 to CXCL5-induced neutrophil chemotaxis.

To visualize neutrophils directly, we generated a Ly6G-GFP knock-in mouse line using the Cas9-associated guide RNA technique. This mouse line carries GFP in the Ly6G (neutrophil marker) allele, enabling visualization of neutrophils by its fluorescence. Intraplantar CXCL5 injection triggered remarkable neutrophil migration to the injected hind paw, as revealed by excessive Ly6g-GFP⁺ cell accumulation. Pharmacologically antagonizing CGRP receptor or SP NK-1 receptor significantly reduced CXCL5-induced neutrophil recruitment (Fig. 7G and H). This observation was further quantified and confirmed by MPO assay (Fig. 7I). Lastly, we tested whether CXCL5 could trigger CGRP and

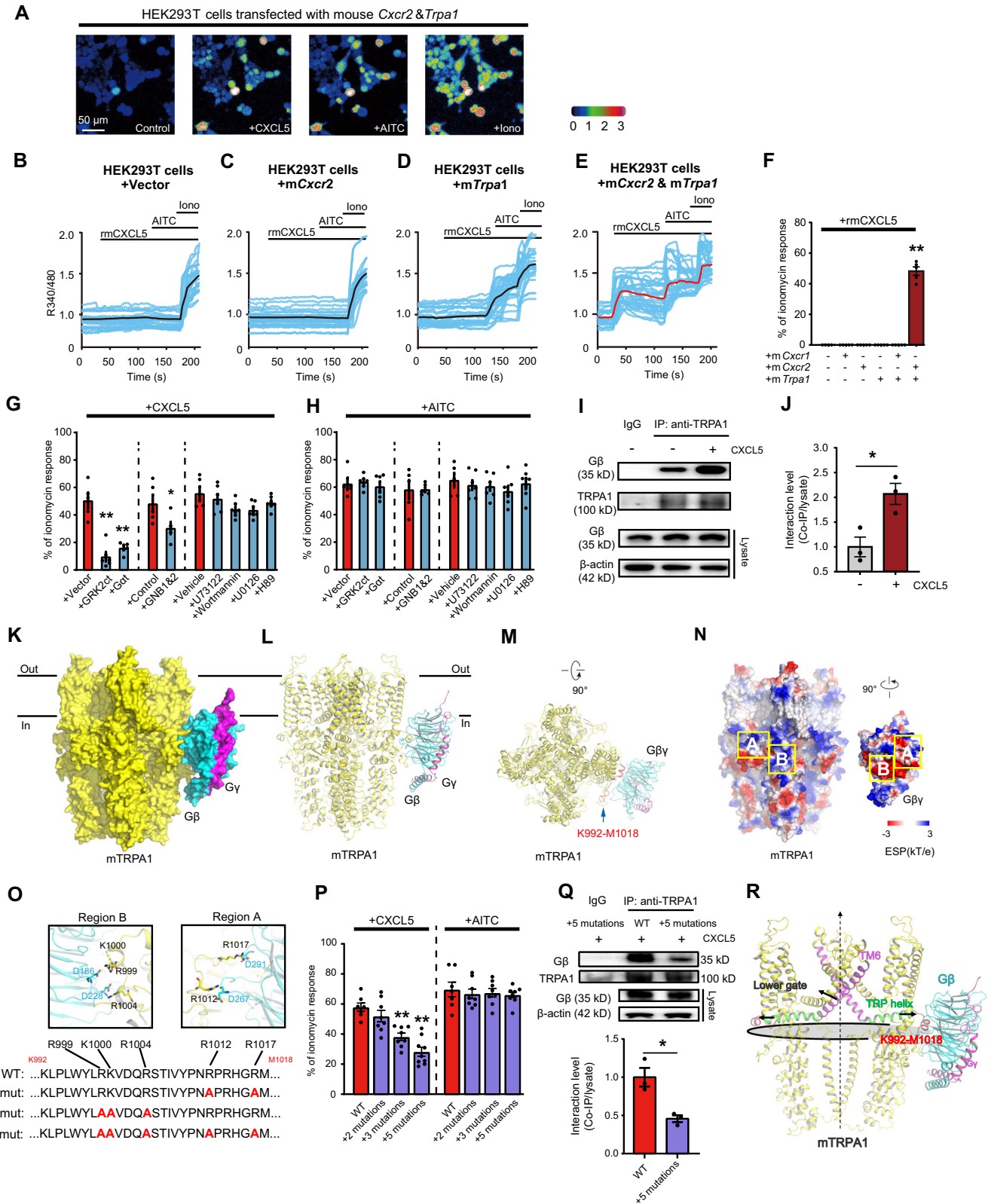

SP release in DRG neuron culture in vitro (Fig. 7J). CXCL5 (30 nM) or high K+ (40 mM) resulted in a significant release of CGRP and SP in the culture medium. Pharmacological blocking of CXCR2 or TRPA1 significantly reduced CXCL5-induced CGRP and SP release in culture medium (Fig. 7K and L). These results indicate that CXCL5 induces pain by activating CXCR2-TRPA1 signaling in nociceptive sensory neurons, which in turn promotes CGRP and SP release and

subsequent plasma extravasation that facilitates CXCL5-induced neutrophil influx.

### Neuronal CXCR2 expression and its coupling with TRPA1 are enhanced in gout arthritis setting

CXCR2 expression in ipsilateral L3-L5 DRG innervating ankle joints was significantly upregulated in gout arthritis model mice vs. control mice

**Fig. 4 | Biochemical basis underlying CXCR2-mediated TRPA1 activation. A** $Ca^{2+}$ imaging of HEK293T cells upon recombinant mouse CXCL5 (rmCXCL5, 100 nM), AITC (100 μM), and ionomycin (10 μM) application. Cells were co-transfected with plasmids containing mouse *Cxcr2* (m*Cxcr2*) and mouse *Trpa1* (m*Trpa1*) genes. Scale bar = 50 μm. **B–E** $Ca^{2+}$ traces evoked by rmCXCL5 in HEK293T cells transfected with vector alone (**B**), m*Cxcr2* (**C**), m*Trpa1* (**D**), or m*Cxcr2*+m*Trpa1* (**E**). **F** Summary of rmCXCL5 evoked $Ca^{2+}$ responses (normalized with ionomycin) in HEK293T cells. **G** Summary of rmCXCL5 evoked $Ca^{2+}$ responses in HEK293T cells transfected with mCXCR2 + mTRPA1. Cells were simultaneously transfected with empty vectors or genes encoding Gαt, GRK2ct, or siRNAs against *Gnb1* and *Gnb2* or treated with U73122 (5 μM), wortmannin (10 nM), U0126 (10 μM), and H89 (10 μM). **H** The same as panel **G**, except AITC was used. **I** and **J** Co-IP showing coupling of Gβ with TRPA1 in HEK293T cells in vehicle- and CXCL5-treated conditions. **K–M** Docking model of mTRPA1 (yellow) with Gβγ (blue and purple). Proteins are shown as surfaces in panel **K**, and proteins in **L** and **M** are shown as secondary structures. Blue arrow indicates the K992-M1018 segment of mTRPA1 for direct binding. **N** Electrostatic surface potential of mTRPA1 and Gβγ proteins. ESP: electrostatic surface potential. **O** Five key residues in K992-M1018 segment of WT channel. Point mutations were illustrated in red. **P** rmCXCL5 or AITC evoked $Ca^{2+}$ responses in HEK293T cells transfected with mCXCR2 and WT mTRPA1 or mutations. **Q** Co-IP showing Gβ coupling with WT or TRPA1 with 5 mutations under CXCL5-treated condition. **R** Proposed working model for Gβγ binding with mTRPA1 and channel opening, Gray oval: Cross-section of the largest part of mTRPA1 radius. One-way ANOVA with Bonferroni's post hoc test was used for panels (**F**, **G**, **H**, and **P**). Student's unpaired *t*-test (two-tailed) was used for panels (**J** and **Q**). *$p < 0.05$, **$p < 0.01$. Data are shown as mean ± SEM. The *n* number, exact *p*-value, and statistical results are provided in Source Data file.

8 and 24 h after model establishment (Fig. 8A). CXCR2 immunostaining in ipsilateral L3-L5 DRG was stronger in model mice than control mice (Fig. 8B and C). We studied the potential coupling of CXCR2 with TRPA1 by Co-IP. Co-IP using anti-CXCR2 antibody revealed coupling of TRPA1 with CXCR2 in DRG under control conditions. The coupling was enhanced in DRG from gout arthritis model mice (Fig. 8D). Vice versa, co-IP using anti-TRPA1 antibody identified similar enhanced coupling of CXCR2 with TRPA1 in gout arthritis condition (Fig. 8E). We then studied CXCL5-induced $Ca^{2+}$ responses in DRG neurons from control and gout model mice. The percentage of CXCL5-responding neurons was significantly increased in the gout model group vs. the control group (Fig. 8F and G). These results demonstrated neuronal CXCR2 expression, its coupling with TRPA1, and its function, are all increased in ankle joint innervating DRG neurons under gout arthritis condition.

### Neuronal CXCR2 contributes to gout arthritis pain, joint inflammation, and gait impairments

In the gout arthritis model, both male and female SNS-Cre *Cxcr2*[fl/fl] mice exhibited significantly improved mechanical allodynia and less ankle diameter increase vs. *Cxcr2*[fl/fl] mice (Fig. 9A and B, Supplementary Fig. 8A and B). Hematoxylin and eosin (H&E) staining revealed a significant increase in ankle joint histopathological score of *Cxcr2*[fl/fl] mice in gout arthritis condition, which was attenuated in SNS-Cre *Cxcr2*[fl/fl] mice (Supplementary Fig. 9A and B). Neutrophil infiltration is a hallmark of acute gout arthritis and contributes to acute pain and joint inflammation[49]. SNS-Cre *Cxcr2*[fl/fl] mice showed significantly reduced neutrophil infiltration in ankle joint vs. *Cxcr2*[fl/fl] mice in gout arthritis condition, revealed by MPO assay and flow cytometry (Fig. 9C–G). CGRP and SP levels were both significantly increased in ankle joints of *Cxcr2*[fl/fl] mice in the gout arthritis context but significantly reduced in SNS-Cre *Cxcr2*[fl/fl] mice (Fig. 9H and I). Pharmacological inhibition of the CGRP receptor or SP NK-1 receptor significantly attenuated the infiltration of neutrophils as well as macrophages in periarticular tissues of model mice (Fig. 9J–L, Supplementary Fig. 10). In addition, MSU induced upregulation of a bunch of cytokines in ankle joints of SNS-Cre negative control mice. SNS-Cre *Cxcr2*[fl/fl] mice showed significantly reduced expressions of *Il1b*, *Il6*, *Tnfa*, *Cxcl2*, *Cxcl1*, *Ccl2* and *Ccl3* (Fig. 9M). Gait analysis indicated model mice showed obvious gait impairment in terms of stance, stride length, and paw area ratios in ipsilateral hind paws 8 h after model establishment. These impairments were reversed in SNS-Cre *Cxcr2*[fl/fl] mice (Fig. 9N). These findings indicated neuronal CXCR2 contributes to gout arthritis pain, joint inflammation, and gait impairments.

### CXCR2 is expressed in human DRG neurons and CXCL5 is elevated in serum of acute gout arthritis patients

In order to know whether our findings may have translational significance, we studied CXCR2 expression in human DRG samples. PCR showed that *CXCR2*, but not *CXCR1* gene, was expressed in human DRG (Fig. 10A). Immunostaining revealed a majority of CXCR2 expressing cells (>95%) was co-stained with NeuN, indicating CXCR2 was mainly expressed in human DRG neurons (Fig. 10B). Applying recombinant human CXCL5 (rhCXCL5, 100 nM) robustly activates $Ca^{2+}$ transient in HEK293T cells co-transfected with both human *TRPA1* and human *CXCR2* (Fig. 10C–G). But rhCXCL5 was not effective in cells transfected with empty vector or human *TRPA1* or human *CXCR2* alone (Fig. 10C–G). We obtained serum samples from acute gout arthritis patients and healthy controls and studied serum CXCL5 levels (Table S3). As expected, plasma uric acid and IL-1β levels in acute gout arthritis patients were significantly higher than healthy controls (Fig. 10H, Table S3). Moreover, serum CXCL5 was significantly elevated in acute gout arthritis patients than healthy controls (Fig. 10I). Spearman's correlation identified positive correlations between CXCL5 and uric acid and IL-1β in serum of acute gout arthritis patients (Fig. 10J–K). Therefore, we identified CXCR2 expression in human DRG neurons and an increased serum CXCL5 level in acute gout arthritis patients, indicating a potential relevance of CXCL5-neuronal CXCR2-TRPA1 signaling to joint pain and inflammation of gouty patients.

## Discussion

CXCL5 is a potent neutrophil chemoattractant by binding with neutrophil CXCR2. An earlier study reported that CXCL5 expression was significantly increased in the skin of UVB-induced inflammatory pain model rat and in UVB-exposed human skin[19]. Neutralizing CXCL5 significantly attenuated mechanical allodynia of UVB-induced inflammatory pain model rats, demonstrating a critical role of CXCL5 in mediating UVB-induced skin pain[19]. CXCL5 injection induces neutrophil and macrophage infiltration. But CXCL5-induced pain occurred much earlier than leukocyte infiltration[19], suggesting CXCL5 might directly act on sensory neurons. However, there is still a lack of evidence supporting this notion. Here we found that bath application of CXCL5 produced obvious $Ca^{2+}$ transients in mouse DRG neurons. Pharmacology combined with genetic methods revealed that CXCL5 acts upon CXCR2 to trigger TRPA1 activation. CXCL5 produced membrane depolarization and hyperexcitability in DRG neurons via CXCR2 and TRPA1-dependent mechanisms. Our results demonstrate that CXCL5 can act upon CXCR2 to trigger TRPA1 activation and produce DRG neuron hyperexcitability.

Previous work suggests Gβγ may serve as downstream signaling of GPCR to activate TRPA1[33,34]. But how Gβγ works to activate TRPA1 remains unknown. Gβγ regulates channel activity via direct binding to the channel or via triggering intracellular signaling pathways[40,41]. We used the HEK293T system and explored the biochemical basis underlying how CXCR2 activates TRPA1. We first confirmed Gβγ's contribution by showing that Gβγ sequestration or Gβ knockdown reduced CXCR2-mediated TRPA1 activation. However, blocking Gβγ-related intracellular signaling has no effect. Co-IP confirmed Gβ's binding to TRPA1 is increased upon CXCR2 activation. This result indicates Gβγ binds with TRPA1 upon CXCR2 activation. We then performed protein-protein docking to predict the mTRPA1 and Gβγ binding model. The

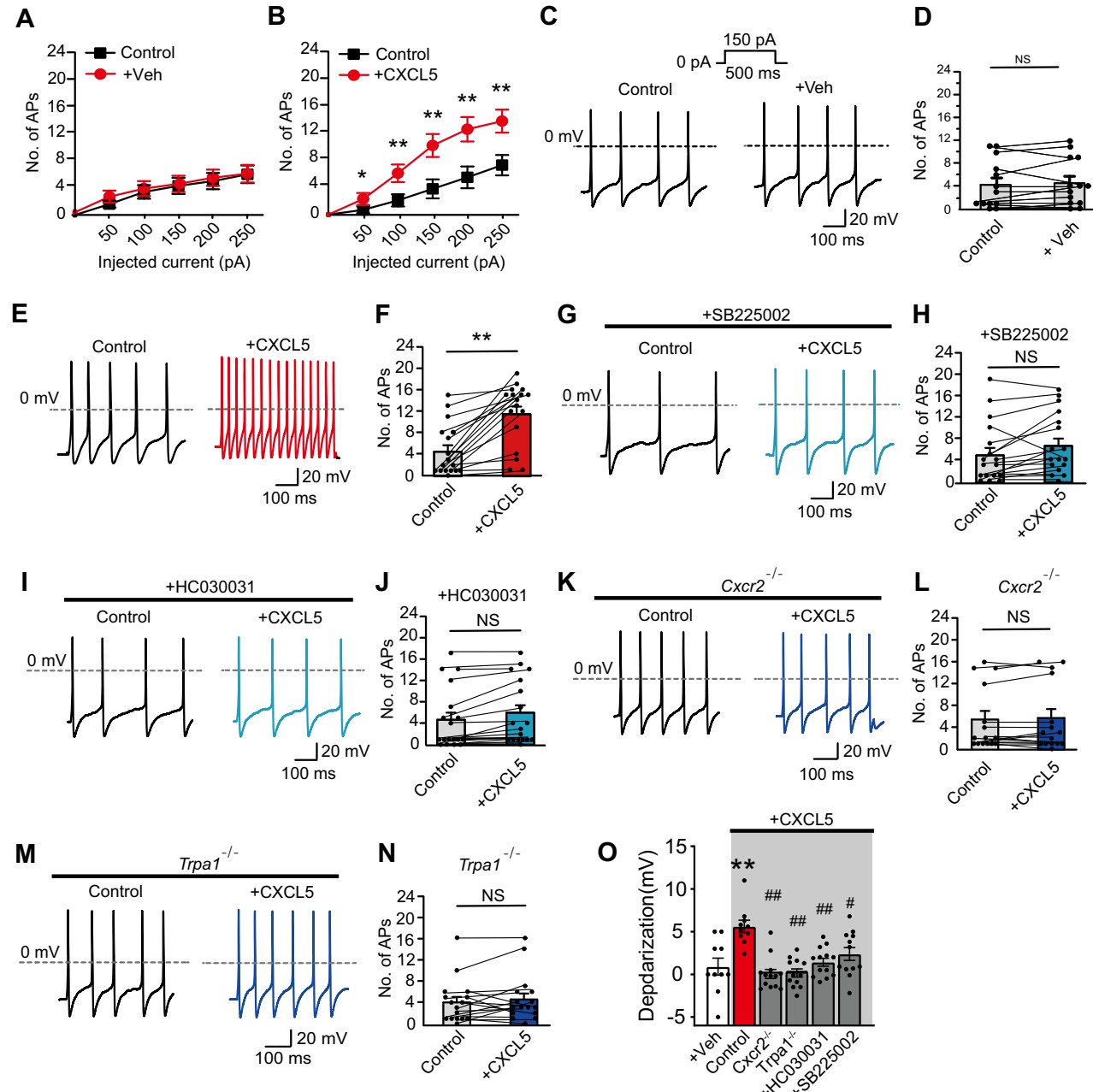

**Fig. 5 | CXCL5 depolarizes and excites mouse nociceptive DRG neurons via CXCR2 and TRPA1-dependent mechanism. A, B** Summary of the No. of action potentials (AP) elicited by injecting different amounts of currents into small-sized DRG neurons (capacitance<42 pF) under the vehicle- (panel **A**) or CXCL5 (100 nM)-treated condition (panel **B**). *$p < 0.05$, **$p < 0.01$ vs. control group. **C** Representative AP firing traces from vehicle (0.1% PBS)-treated DRG neuron. **D** Summary of the number of APs triggered by a vehicle. **E** Representative AP firing traces recorded from CXCL5-challenged DRG neuron. **F** Summary of the No. of APs triggered by CXCL5. **G, I** Representative AP firing traces from CXCL5-challenged DRG neuron pretreated with SB225002 (100 nM, panel **G**) or HC030031 (100 μM, panel **I**). **H** and

**J** Summary of the number of APs triggered by CXCL5 under SB225002 (panel **H**) or HC030031 (panel **J**) pretreated condition. **K, M** Representative AP firing traces triggered by CXCL5 from *Cxcr2*⁻/⁻ (panel **K**) and *Trpa1*⁻/⁻ (panel **M**) mice. **L** and **N** Summary of the No. of APs triggered by CXCL5 from *Cxcr2*⁻/⁻ (panel **L**) and *Trpa1*⁻/⁻ (panel **N**) mice. **O** Summary of depolarization of membrane potential in DRG neurons caused by CXCL5 in different groups. **$p < 0.01$ vs. Veh group. ##$p < 0.01$ vs. control group. NS: no significance. One-way ANOVA with Bonferroni's post hoc test for panel (**O**). Student's paired *t*-test (two-tailed) was for others. Data are shown as mean ± SEM. The *n* number, exact *p*-value, and statistical results are provided in the Source Data file.

region we predicted where TRPA1 binds to Gβ is located at the C-terminal end of TRP helix[42]. We further identified five key residues located in the K992-M1018 segment of mTRPA1 that show strong electrostatic interaction with Gβ. CXCR2-mediated TRPA1 activation was significantly reduced after mutating these five residues. The mutated channels were functional since AITC evoked similar responses among WT and mutated ones. Finally, Co-IP confirmed that the binding of TRPA1 with Gβ was markedly reduced by mutations of all five

residues. We speculate that the binding of Gβ to the K992-M1018 segment of TRPA1 may cause outward movement of the TRP helix, which in turn pulls TM6 to undergo metastasis and thus opens the lower gate of the TRPA1 channel. But it should be noted that CXCR2-induced TRPA1 activation was not completed eliminated by these mutations we identified. It remains likely that some other regions may also contribute to Gβ-binding to TRPA1 and subsequent channel activation. Moreover, protein–protein docking cannot fully map the full

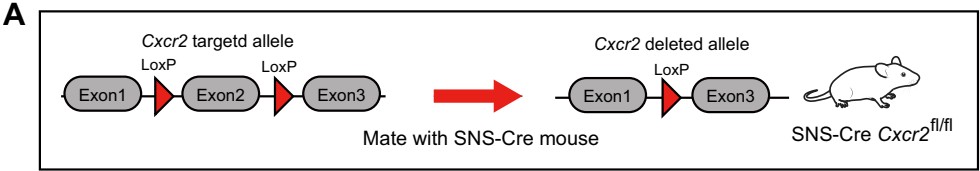

**Fig. 6 | CXCR2 expressed in nociceptive DRG neurons contributes to CXCL5-induced mechanical allodynia and joint inflammation in mice. A** Schematic picture showing the overall design of *Cxcr2* conditional knockout mice in nociceptive DRG neurons using SNS-Cre mouse line. **B** Immunostaining showing CXCR2 expression in DRG and spleen of Cxcr2^fl/fl and SNS-Cre *Cxcr2*^fl/fl mice. Scale bars, 50 μm (DRG) and 100 μm (Spleen). **C** Summary of % of CXCR2 positively expressing (CXCR2⁺) cells in DRG. **D** Summary of CXCR2 fluorescence intensity/field in the spleen. **E** Western blot showing CXCR2 expression in DRG (left) and SCDH (right) of *Cxcr2*^fl/fl and SNS-Cre *Cxcr2*^fl/fl mice. **F, G** Comparisons of CXCL5-induced mechanical

allodynia and ankle diameter between *Cxcr2*^fl/fl and SNS-Cre *Cxcr2*^fl/fl mice. CXCL5 (300 ng/site) was injected into the ankle joint of mice after the baseline test (0 h). **H–K** qPCR showing inflammation-related gene expressions, including *Il1b, Tnfa, Il6,* and *Ifng*. *$p < 0.05$, **$p < 0.01$. Two-way ANOVA (repeated measures) with Bonferroni's post hoc test was used (panels **F** and **G**). One-way ANOVA with Bonferroni's post hoc test for panels (**H–K**). Student's unpaired *t*-test (two-tailed) was for others. Data are shown as mean ± SEM. The *n* number, exact *p*-value, and statistical results are provided as a Source Data file. The mouse schematic picture in panel **A** was drawn using SciDraw (https://scidraw.io/).

---

details of protein interactions; therefore, microscopic mechanisms of protein interactions will need to be further addressed via the cryo-electron microscopy technique.

It is traditionally believed that CXCL5 produces chemotaxis on neutrophils via CXCR2 expressed on neutrophils[50]. Here we identified a previously unidentified role of CXCR2 expressed in nociceptor neurons in CXCL5-induced neutrophil chemotaxis by specifically deleting *Cxcr2* in nociceptive sensory neurons. The validity of this approach is demonstrated by specific knockdown of CXCR2 expression in sensory

neurons but not in neutrophils or other tissues. With the aid of this mouse strain, we found intraplantar CXCL5 injection elicited obvious neurogenic inflammation and neutrophil recruitment that depends on neuronal CXCR2. CXCL5-induced neurogenic inflammation also depends on TRPA1 and can be blocked by CGRP and substance P receptor antagonists. We found CXCR2 largely co-expressed with CGRP⁺ DRG neurons, suggesting CXCL5 may activate neuronal CXCR2 to trigger neuropeptide release. Indeed, we found that CXCL5 can activate neuronal CXCR2 and trigger CGRP and SP release in hind paw

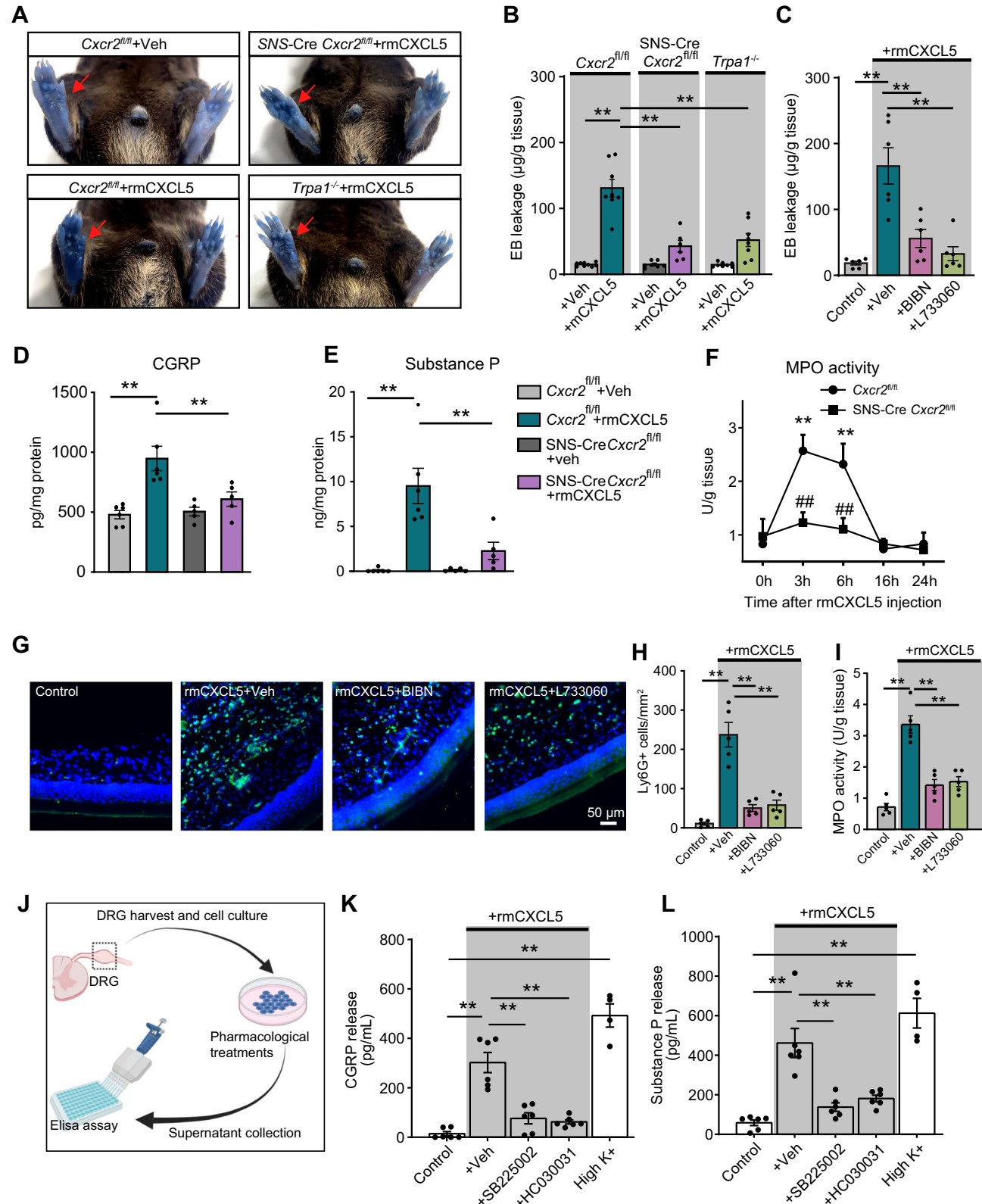

tissues and DRG neurons. CGRP and SP are neuropeptides that can produce strong vasodilation and plasma extravasation by acting on CGRP and SP NK1 receptors expressed on blood vessels[51,52]. Pharmacological blocking of CGRP and SP NK1 receptors significantly reduced CXCL5-induced neutrophil recruitment. Therefore, these findings unraveled a previously unidentified contribution of CXCR2 in nociceptive sensory neurons to CXCL5-induced neutrophil chemotaxis.

Neutrophil infiltration is a hallmark of acute gout arthritis[49,53]. Infiltrated neutrophils produce large amounts of ROS and release inflammatory mediators that contribute to gout arthritis pain and inflammation[10,14]. MSU-induced neutrophil influx occurred through a CXCR2-dependent manner[15]. In light of the findings we obtained from CXCL5, we proceeded to examine the contribution of neuronal CXCR2 to neutrophil infiltration in gout arthritis. We found neutrophil

**Fig. 7 | Neuronal CXCR2 contributes to CXCL5-induced neurogenic inflammation and neutrophil chemotaxis in mice. A** Representative photos showing plasma extravasation of EB 3 h after CXCL5 injection into hind paws of WT, SNS-Cre *Cxcr2*[fl/fl], and *Trpa1*[−/−] mice. **B, C** Quantification of EB intensity 3 h after CXCL5 (300 ng/site) injection into hind paws of WT, SNS-Cre *Cxcr2*[fl/fl], *Trpa1*[−/−] mice, or mice treated with BIBN4096BS (BIBN) or L-733060. **D, E** ELISA of CGRP and SP levels in hind paw tissues. **F** Time course showing MPO results that measure neutrophil accumulation in hind paw tissues upon CXCL5 injection. *$p < 0.05$, **$p < 0.01$ vs. 0 h time point of *Cxcr2*[fl/fl] group, #$p < 0.05$ vs. *Cxcr2*[fl/fl] group. **G** Neutrophil recruitments in hind paws revealed by Ly6G-GFP fluorescence signals in transgenic mice 3 h after CXCL5 injection. BIBN (60 ng/site) or L-733060 (1 μg/site) was co-

applied (i.pl.) with CXCL5 to block CGRP receptor or SP NK-1 receptor. Purple indicates DAPI staining. Scale bar = 50 μm. **H** Summary of the panel (**G**). **I** MPO activity assays of hind paw tissues from 4 groups of mice 3 h after CXCL5 injection. **J** Cartoon showing the workflow of CGRP and SP ELISA from DRG neuron culture. (**K&L**) CGRP and SP levels were measured in the supernatant of DRG neuron culture medium after treatments by vehicle (0.1% DMSO), SB225002 (100 nM), HC030031 (50 μM), and high K⁺ (40 mM). *$p < 0.05$, **$p < 0.01$. Two-way ANOVA (repeated measures) with Bonferroni's post hoc test was used in panels (**F**). One-way ANOVA with Bonferroni's post hoc test for others. Data are shown as mean ± SEM. The *n* number, exact *p*-value, and statistical results are provided as a Source Data file. The schematic picture in (**J**) was created with Biorender.

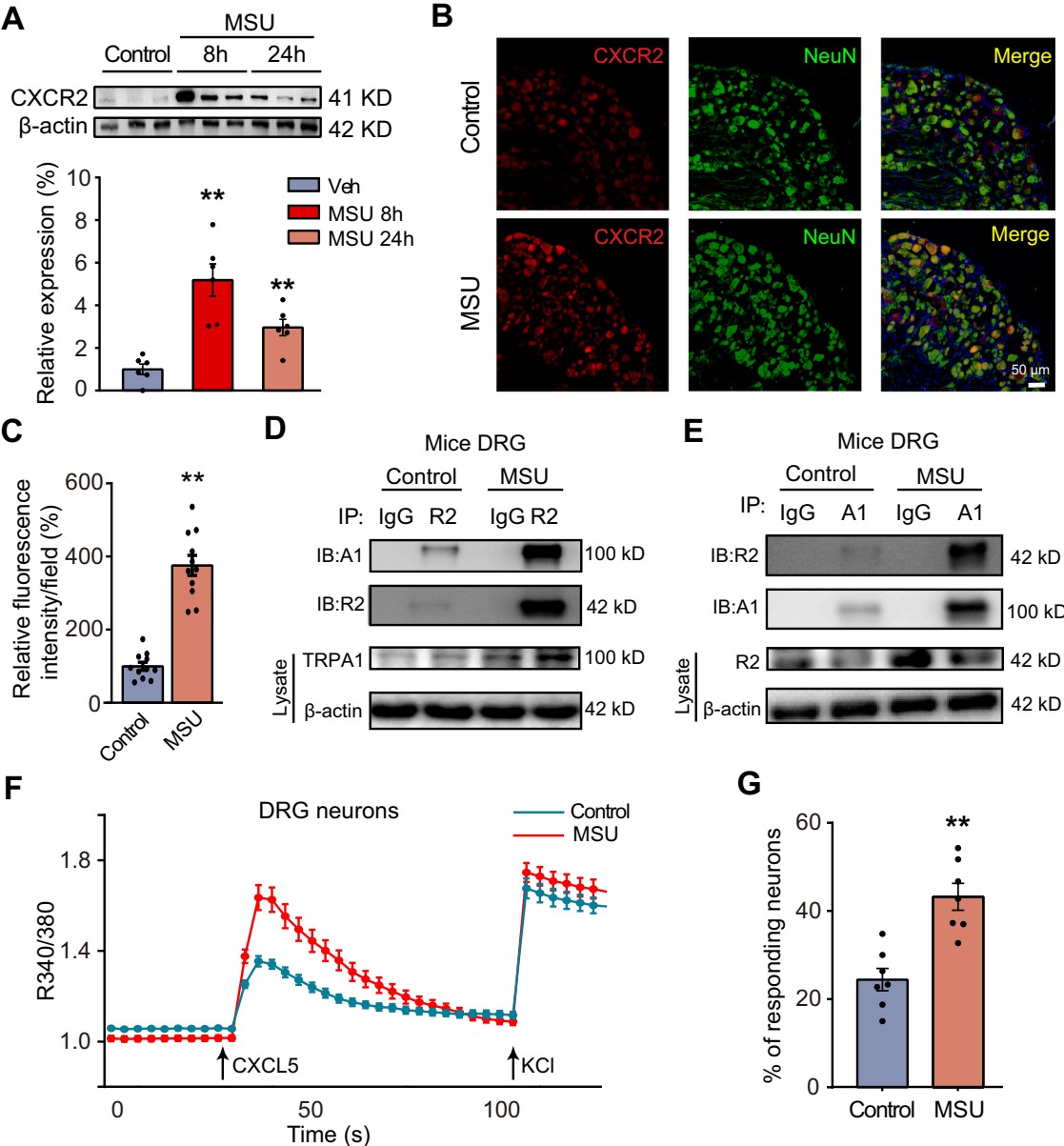

**Fig. 8 | CXCR2 expression and its function in mouse DRG neurons are enhanced in gout arthritis conditions. A** CXCR2 expression in ipsilateral L3−L5 DRG neurons of the control group (vehicle-treated) and MSU group was examined by Western blot. Upper panel shows representative gel and the lower panel shows summarized data. *n* = 6 mice/group. **B** Representative CXCR2 expression in ipsilateral L3−L5 DRG neurons of control and MSU groups by immunostaining. Scale bar = 50 μm. **C** Summary of relative fluorescence intensity of CXCR2 immunostaining. Control group value was taken as 100%. *n* = 11 mice/group. **D, E** Co-IP showed the coupling of CXCR2 with TRPA1 and vice versa in mice DRG in control and MSU-treated

conditions. **F** Ca²⁺ imaging showing CXCL5-induced Ca²⁺ responses in ipsilateral L3−L5 DRG neurons of control and MSU groups. CXCL5 and KCl were applied at time points shown by the arrows. **G** Summary of the % of responding neurons challenged by CXCL5. *n* = 7 tests/group. Neurons were obtained from 5 mice in each group. **$p < 0.01$. One-way ANOVA with Bonferroni's post hoc test for panel (**A**). Student's unpaired *t*-test (two-tailed) for panels (**C** and **G**). The data are shown as mean ± SEM. The *n* number, exact *p* value, and statistical results are provided in the Source Data file.

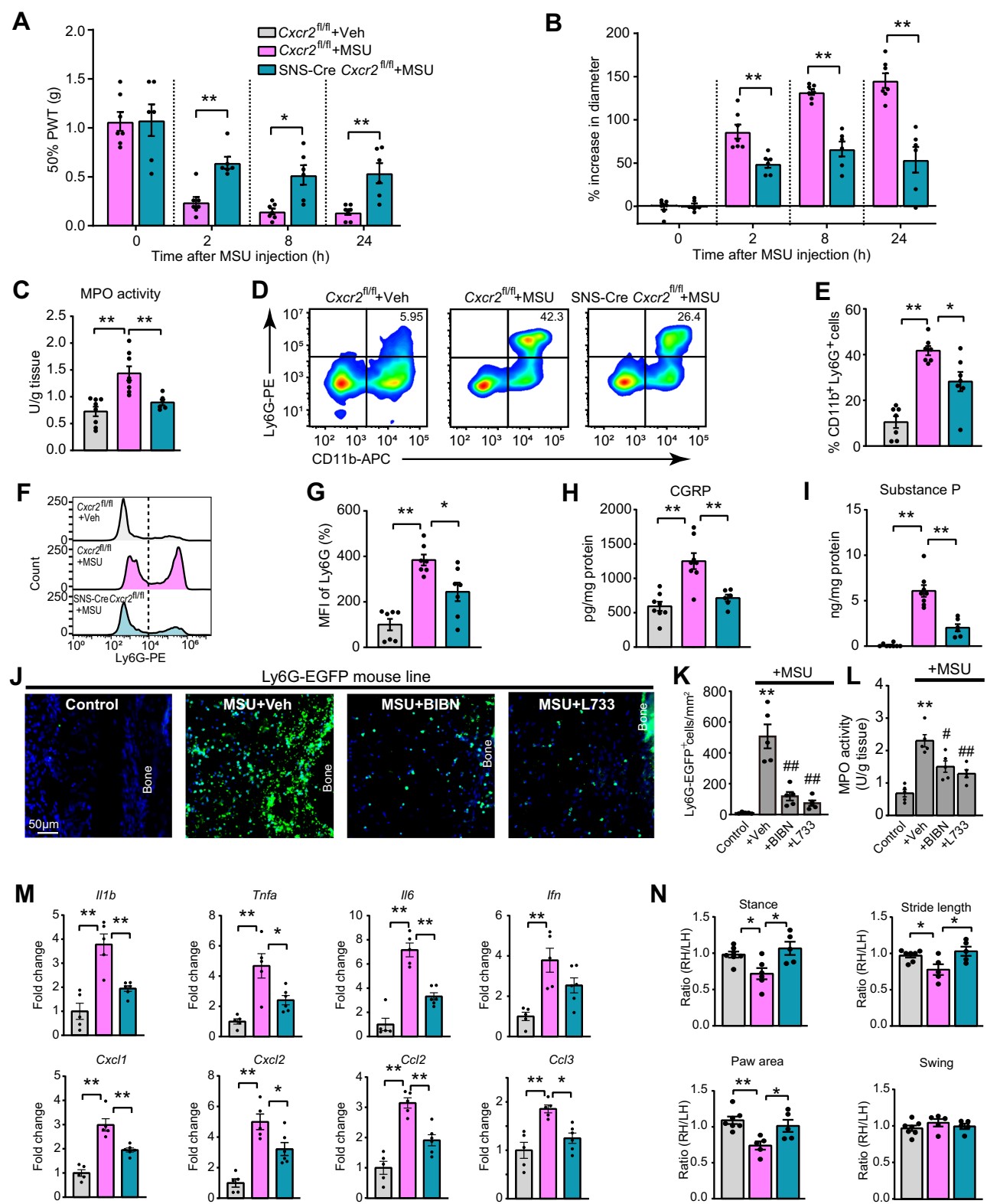

infiltration was significantly reduced in SNS-Cre *Cxcr2*<sup>fl/fl</sup> mice. CGRP and SP were significantly increased in the ankle joints of gout model mice. More importantly, the increase in CGRP and SP depends on neuronal CXCR2. Pharmacological antagonizing CGRP or NK1 receptor strongly reduced neutrophil infiltration in ankle joints of gout arthritis model mice. Therefore, these results demonstrate that neuronal CXCR2 activation results in CGRP and SP release, which triggers neurogenic inflammation and plasma extravasation and thereby

contributes to neutrophil infiltration in gout arthritis. Our study demonstrates a critical contribution of CXCR2 expressed in nociceptor neurons to neutrophil infiltration in gout arthritis.

CXCR2 expression was upregulated in DRG neurons in gout arthritis conditions. The coupling of CXCR2 with TRPA1 or vice versa was significantly enhanced in DRG neurons of gout arthritis model mice. CXCL5-induced Ca$^{2+}$ response was increased in DRG neurons isolated from gout arthritis model mice vs. control mice. These results

**Fig. 9 | Neuronal CXCR2 contributes to mechanical allodynia, joint inflammation, and gait impairments in gout arthritis model mice. A, B** Changes in 50% PWT (**A**) and ankle joint diameter (**B**) of *Cxcr2*^fl/fl and SNS-Cre *Cxcr2*^fl/fl group of mice upon gout arthritis model establishment. **C** Summary of MPO activity assay. **D** Representative FACS plots of neutrophils (CD11b⁺Ly6G⁺) in ankle joint articular cavity from Cxcr2^fl/fl and SNS-Cre *Cxcr2*^fl/fl mice 8 h after intraarticular MSU/Veh injection. **E** Summarized % of neutrophils (CD11b⁺Ly6G⁺) within gated cell populations. **F** Representative histograms comparing the mean fluorescence intensity (MFI) of Ly6G expression on gated cells from *Cxcr2*^fl/fl+Veh (Top), *Cxcr2*^fl/fl + MSU (mid), or SNS-Cre *Cxcr2*^fl/fl + MSU (Bottom) group of mice. **G** Quantification of the results from panel (**F**). **H, I** CGRP and SP in ankle joint tissues of three groups of

mice. **J** Representative photos of Ly6G-GFP signals in periarticular tissues of the ankle joint. DAPI was in purple. Scale bar = 50 μm. **K** Summary of the number of Ly6G-GFP⁺ cells/mm² in four groups. BIBN (60 ng/site) or L-733060 (1 μg/site) was co-applied (i.a.) with MSU. **L** MPO activity assays of ankle joints from four groups. **M** Gene expression of certain key inflammatory cytokines or chemokines in ankle joints. **N** Summarized gait parameters, including ration (RH/LH) of stance, paw area, stride length, and swing. *$p < 0.05$, **$p < 0.01$. Two-way ANOVA (repeated measures) with Bonferroni's post hoc test for panels (**A** and **B**). One-way ANOVA with Bonferroni's post hoc test for others. Data are shown as mean ± SEM. The $n$ number, exact $p$-value, and statistical results are provided in the Source Data file.

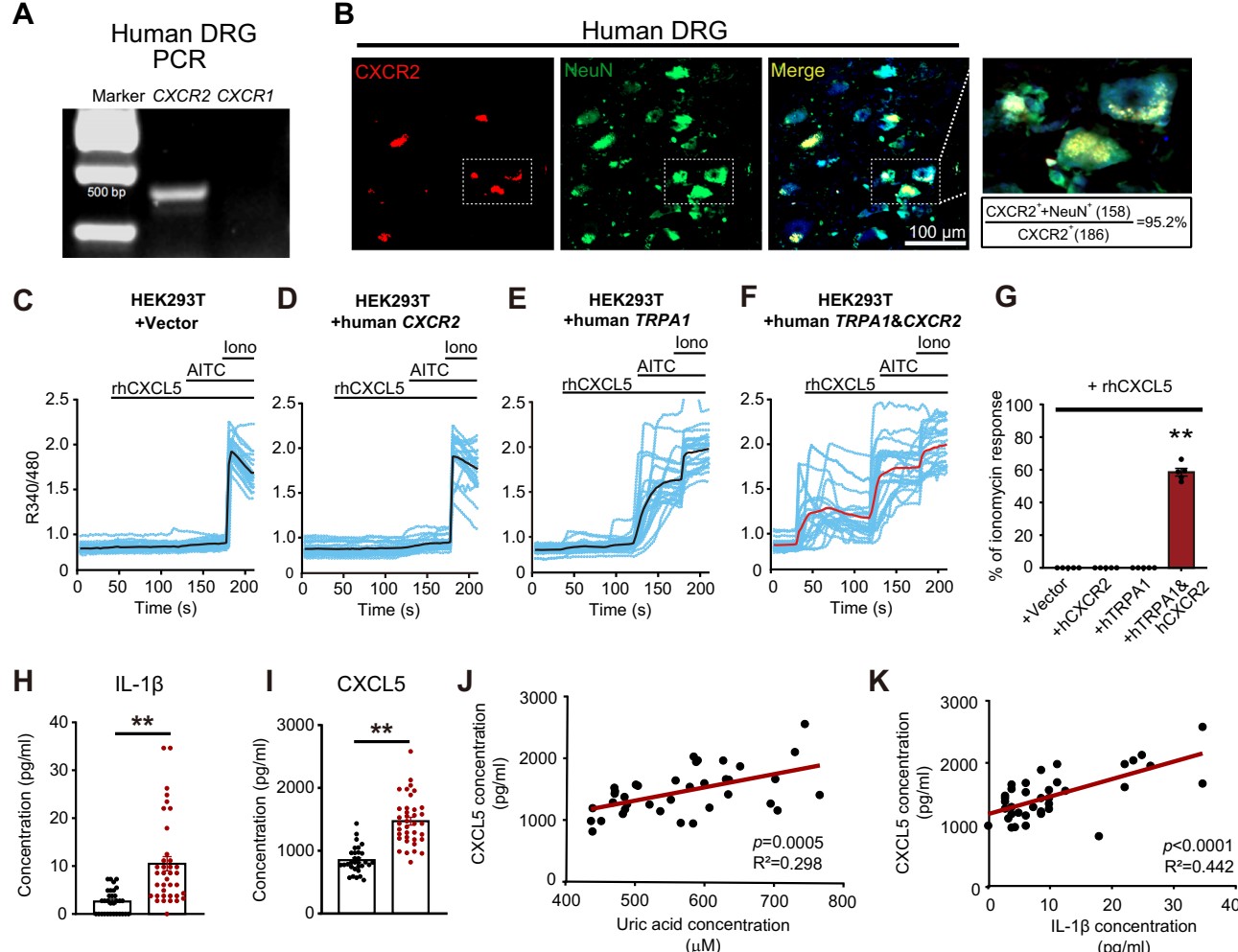

**Fig. 10 | CXCR2 is expressed in human DRG neurons and CXCL5 is elevated in serum of acute gout arthritis patients. A** *CXCR1* and *CXCR2* gene expression in human DRG by PCR. The experiment has been repeated three times.
**B** Immunostaining showing co-expression of CXCR2 with NeuN in human DRG. Scale bar = 100 μm. White dashed box area is further enlarged and shown in the right panel. The data is summarized from six observational fields. **C**−**F** Ca²⁺ imaging using HEK293T cells transfected with plasmids containing empty vector (**C**), human *CXCR2* (**D**), human *TRPA1* (**E**), or human *CXCR2*+human *TRPA1* (**F**). Recombinant human CXCL5 (rhCXCL5, 100 nM) and AITC (100 μM) were used to challenge the

cells. Ionomycin (10 μM) was applied at the end to light up all live cells. **G** Summary of the effects of rhCXCL5 on HEK293T cells. **$p < 0.01$. **H, I** Serum IL-1β and CXCL5 levels in healthy control and gout arthritis patients. **$p < 0.01$. **J** and **K** Correlation of serum CXCL5 level with serum levels of uric acid or IL-1β from gout arthritis patients. One-way ANOVA with Bonferroni's post hoc test for panel (**G**). Student's unpaired $t$-test (two-tailed) for panels (**I** and **J**). Linear regression analysis for panels (**J** and **K**). Data are shown as mean ± SEM. The $n$ number, exact $p$-value, and statistical results are provided as a Source Data file.

indicate that CXCL5 produces stronger TRPA1 activation via neuronal CXCR2 in DRG neurons in gout arthritis. This finding indicates that CXCL5 released during gout arthritis may produce more pain and neurogenic inflammation by acting upon the upregulated neuronal CXCR2. However, the mechanisms underlying the upregulation of CXCL5-CXCR2 signaling await further investigation. It is possible that

the activation of a signaling cascade by certain inflammatory mediators may promote neuronal CXCR2 overexpression in gout arthritis.

Here we found that CXCL5 is significantly elevated in the serum of gout arthritis patients. Serum level of CXCL5 positively correlates with urate and certain classical inflammation markers, e.g., IL-1β. We also identified CXCR2 expression in human DRG neurons. Using

HEK293T cells as a heterologous expression system, we showed that recombinant human CXCL5 protein triggers robust human TRPA1 activation via acting on human CXCR2. These findings demonstrate the clinical and translational significance of our study. *Cxcr2* global knockout mice also showed significantly reduced mechanical allodynia and joint inflammation in gout arthritis settings. However, it should be noted that *Cxcr2* global knockout mice exhibit obvious deficits in wound healing due to decreased neovascularization[54]. Thus, the neuronal-specific *Cxcr2* knockdown strategy may represent a plausible approach for gout arthritis management with more specificity on joint innervating sensory neurons.

We found that neutralizing CXCL1 or CXCL2 only showed ameliorating effects on ankle joint inflammation but not on joint pain, whereas neutralizing CXCL5 ameliorates both pain and ankle inflammation of gout arthritis model mice. CXCL1, 2, and 5 bind with CXCR2 to produce biological effects. We used $Ca^{2+}$ imaging to further compare their abilities to excite DRG neurons. We found that among these chemokines, CXCL5 produced the most obvious DRG neuron activation, which was significantly stronger than CXCL1. In contrast, CXCL2 produced no activation at all. We further confirmed that both CXCL5 and CXCL1's activating effects on DRG neurons were CXCR2-dependent. This finding implies that CXCL5 can be more powerful than CXCL1 or CXCL2 in producing nociceptive signals during gout arthritis. RNA-Seq further reveals that CXCL5 expression is the most upregulated among these chemokines in the ankle joints of model mice. Regarding the relatively high expression of CXCL5 in the inflamed ankle joints and its high potency in exciting sensory neurons, neutralization of CXCL5 can produce the most effective ameliorating effect on both pain and inflammation in gout model mice. Further studies will be needed to thoroughly investigate the mechanisms underlying different potencies of these chemokines to activate neuronal CXCR2.

We observed some residual pain responses still remained in SNS-Cre *Cxcr2*^fl/fl mice. This could be due to the complex nature of the inflamed joint of gout arthritis. During inflammation, multiple pro-inflammatory cytokines or chemokines are released to contribute to the overall pain and inflammation of gout arthritis. For example, we found that IL-33, a pro-inflammatory cytokine, is significantly increased in the inflamed ankle joint and contributes to gout arthritis pain by promoting neutrophil-dependent ROS production and TRPA1 channel activation through receptor ST2[14]. We also found that IL-1β generated via NLRP3 inflammasome during gout arthritis contributes to gout arthritis pain by promoting TRPV1 channel overexpression in sensory neurons[24,55]. Besides, oxidative stress generated during gout arthritis can directly activate TRPA1 in sensory neurons to trigger pain in gout model mice[56]. One recent study also highlights an important role of macrophage TRPV4's involvement in gout pain and inflammation[11]. Here we also observed *Il6* as the highest upregulated cytokine gene. IL-6 can promote functional expression of TRPV1 in DRG neurons and contributes to peripheral sensitization[57]. It is likely that IL-6-mediated signaling may also contribute to gout pain. Therefore, there are many other potential signaling pathways involved in gout arthritis pain, reflecting the complexity of gout arthritis pain mechanisms.

We propose that CXCL5 released in the joint during gout arthritis acts on CXCR2 expressed in joint-innervating nociceptive sensory neurons to trigger TRPA1 activation via Gβγ signaling, resulting in membrane depolarization and hyperexcitability that produce pain signals. CXCR2's expression and coupling with TRPA1 are increased under gout arthritis conditions. TRPA1 activation in peptidergic nociceptors further triggers $Ca^{2+}$ influx that facilitates neuropeptide CGRP and SP release. These neuropeptides act on the CGRP receptor and SP NK1 receptor in nearby blood vessels to trigger potent vasodilation and plasma extravasation that facilitate CXCL5-induced neutrophil chemotaxis from blood vessels to the inflammatory site. The infiltrated

neutrophils release ROS and inflammatory mediators that further contribute to gout pain and joint inflammation[14,15]. Therefore, we unraveled the critical role of the CXCL5-neuronal CXCR2–TRPA1 axis in joint-innervating nociceptive sensory neurons to drive acute gout arthritis pain and joint inflammation (Supplementary Fig. 11). We propose that specific targeting CXCR2 in joint-innervating sensory neurons could be a potential strategy for gout arthritis management in the future.

## Methods

### Study approval
All animal uses and procedures were applied according to protocols approved by the Laboratory Animal Management and Welfare Ethical Review Committee of Zhejiang Chinese Medical University (Permission No. #IACUC-20190819-04). All efforts were made to minimize the number and suffering of the animals used. The collection of clinical samples was conducted following the Declaration of Helsinki and approved by the Ethics Committee of the Second Affiliated Hospital of Hebei Medical University (#2022-R282). The serum samples were derived from leftover samples from diagnostic laboratories with sufficiently small harm risks to the patients, and the study is observational; thus, the Ethics Committee of the Second Affiliated Hospital of Hebei Medical University waived the requirement for informed consent. The collection of Human postmortem DRG samples was conducted with the approval of the Ethics Committee of Zhejiang Chinese Medical University (#ZJ-2161934-1).

### Animals
C57BL/6 (6–8 weeks, male) was purchased from Shanghai Laboratory Animal Center, Chinese Academy of Sciences. *Cxcr2*^−/− and *Cxcr2*^fl/fl mice were custom-generated by GemPharmatech (Nanjing, China). Ly6g-IRES-GFP knock-in mice were custom-generated via the Cas9-associated guide RNA (gRNA) technique by Shanghai Model Organisms Center, Inc. (Shanghai, China). *Trpv1*^−/− and *Trpa1*^−/− mice were provided by Prof. Zhen-zhong Xu (Zhejiang University School of Medicine). SNS-Cre mice (C57BL/6) were provided by Professor Xu Zhang (Chinese Academy of Sciences, Shanghai, China). *Cxcr2*^fl/fl mice were crossed with SNS-Cre mice to generate SNS-Cre *Cxcr2*^fl/fl mice (male and female). *Cxcr2*^fl/fl mice (male and female) were used as negative controls. The sequences of the primers used for mouse genotyping can be found in Table S1. Animals were housed in the Laboratory Animal Center of Zhejiang Chinese Medical University. Food and water were available ad libitum. The mice were maintained in a pathogen-free environment on a 12-h light/dark cycle at controlled temperature (24 ± 2 °C) and humidity (50–60%). The food was standard mouse chow (containing protein 18%, fat 5%, and fiber 5%). The reporting of animal experiments abides by the ARRIVE guideline. All efforts were made to minimize animal suffering during the experiments. $CO_2$ inhalation was used for the euthanasia of animals.

### Gouty arthritis patient's serum and human postmortem DRG collection
The patients were diagnosed with gouty arthritis based on American College of Rheumatology classification criteria[58] and met the following criteria: (1) 20–60 years old male; (2) have been admitted within 72 h after the onset of an acute gout attack; (3) serum uric level ≥ 420 μM. 37 patients were included in this study. Gouty arthritis is a male pre-dominant disease. Female patients are difficult to recruit in clinics within the time frame. Therefore, only male patients were included. These patients had no other inflammatory conditions and had not taken any anti-inflammatory drugs during the period. 31 age-matched males were included as healthy controls. The serum samples were collected in EDTA tubes and centrifuged (2000 × g, 10 min), and the supernatant was kept under −20 °C for further analysis. All samples were used and analyzed anonymously. The clinical characteristics of

gout arthritis patients and healthy controls are provided in Table S2. Human postmortem DRG samples were collected from body donations to the Department of Human Anatomy, Zhejiang Chinese Medical University.

## Establishment of MSU-induced gouty arthritis mouse model

The gout arthritis model was established through intra-articular (i.a.) injection of MSU crystals (500 µg) suspended in endotoxin-free PBS (20 µl) into ankle joints of mice under isoflurane anesthesia as described before[23,24]. Control mice were injected (i.a.) with an equal volume of sterile PBS. Successful establishment of the model was determined by the development of mechanical allodynia and ankle swelling 2 h after MSU injection, as reported[56].

## Animal behavioral test and ankle diameter evaluation

**Mechanical allodynia.** Mice were individually placed in a transparent observation chamber on an elevated metal mesh floor and habituated for at least 30 min before the test. The von Frey filaments were applied perpendicularly to the heel area of the ipsilateral hind paw as described in our recent study[14,59]. The "up-down" testing paradigm was used to determine the paw withdrawal threshold (PWT) and the 50% paw withdrawal threshold (PWT) was calculated by the nonparametric Dixon test.

**Open field test**. 20 lx illumination was applied to the open field. The mice were placed in the testing room 30 min before the test to be accustomed to the environment. The mice were then gently placed in the center of the open field. After 30 s, the ANY-maze software (Stoelting, USA) was started to record the activity of the mice and the data was analyzed afterwards offline.

**Gait analysis.** The gait of mice was recorded and analyzed via the DigiGait imaging system (MouseSpecifics Inc., USA) as reported[55]. The mouse was put on a flat and transparent treadmill which was operated at a constant speed (18 cm/s). A video camera was located underneath the treadmill to record the gait of mice while running. The animals ran on the treadmill for 20 s, and five consecutive strides were averaged per animal and used for gait analysis. Parameters including paw area, swing, stance and stride length were calculated for each mouse through the software and then analyzed.

**Ankle diameter.** The diameter of the ankle joint was determined as an increase in ankle diameter, measured with a digital caliper and was calculated as the difference between the basal value and the test value[24]. The results were shown as % increase in ankle diameter ($D$) and calculated as follows: % increase in ankle diameter = $(D_{after} - D_{before})/D_{before}$. All behavioral tests and ankle joint evaluation were performed by an experimenter blinded to groupings.

## RNA-sequencing and bioinformatics analysis

Mice were euthanized 8 or 24 h post-injection of MSU or vehicle. The injected ankle joints were collected, diced, and stored in RNAlater (Thermal Fisher Scientific, USA). Total RNA from the vehicle and MSU group was extracted using a Trizol reagent (Thermal Fisher Scientific, USA). RNA quality and purity were tested by TapeStation (Agilent, USA) and NanoDrop (Thermo Fisher Scientific, USA). Only RNA samples showing RNA Integrity Number (RIN) ≥ 8.0 and A260/230 ≥ 1.5 were used for RNA-Seq. The samples were sequenced by BGISEQ-500 from BGI Group (Shenzhen, China). Raw fastq files were aligned to the mus musculus mm10 reference genome for mouse data using the STAR 2.5 version aligner. After reads were mapped to mm10, appropriate file conversions (from BAM format to BED format) were made using bedtools v2.17.0. Gene-counts matrix was generated using Rsubread version 1.30.9. Count files were then normalized and analyzed via DESeq2 and custom R scripts. Differential expression was determined using DESeq2. Significance cutoffs were determined as follows: adjust $p$-value (FDR) < 0.001 and the absolute value of |log2 (Fold Change)| > 1.0. Pathway analysis of differentially expressed genes (DEGs) was conducted using DAVID[60]. The RNA-Seq dataset of 24 h time point was derived from our recent study[14] and reanalyzed according to the criteria described above. All raw fastq files and the expression count matrix have been deposited into the National Center for Biotechnology Information's Gene Expression Omnibus (accession number GSE242872).

## Immunofluorescence

The immunofluorescence was performed using standard protocol[61,62]. The samples were first fixed using paraformaldehyde (4%) and transferred to sucrose gradients for 3 days to dehydrate. For ankle joints, decalcification with EDTA was first performed. After dehydration, the specimens were cut into sections with a thickness of 15 µm with a cryostat (CryoStar, NX50, Thermo Fisher, USA). The cut sections were then mounted to glass slides pre-coated with gelatin. 1% BSA + 10% donkey serum was used to block the sections for 2 h at room temperature and then incubated with primary antibodies (Table S1) diluted in blocking solution overnight at 4 °C. Sections were then washed three times for 5 min in PBS. The tissues were then incubated with the secondary antibodies (Donkey anti-rabbit Cy3, Donkey anti-rabbit FITC, Donkey anti-mouse Cy3, and Alexa Fluor 647 Donkey anti-chicken IgG, Table S1) for 1 h at room temperature. The stained and mounted sections were then examined with a Zeiss Apotome 3 microscope (Zeiss, Germany). The images were captured using ZEN software (Zeiss, Germany). To avoid unspecific stained tissues, a cell is judged as a Ly6G or Iba-1 positively labeled cell by simultaneously bearing 2 features: (1) stained positive for Ly6G or Iba-1; (2) having a definite DAPI-labeled nucleus. For quantification, at least three sections from each mouse were selected and averaged as described[63,64]. The fluorescence intensity of 1024 × 1024-pixel resolution images was measured using Image J (NIH, USA).

## HEK293T cell culture and transfection

HEK293T cells were purchased from ATCC (#CRL-3216) and cultured in DMEM supplemented with 10% FBS, 2 mM L-glutamine, 100 U/ml penicillin, and 100 µg/ml streptomycin. The culture was maintained with 5% $CO_2$ in a 37 °C incubator. The siRNA specific to Gβ1 (UACGACGACUUCAACUGCA), Gβ2 (ACGACGACUUCAACUGCAA), and silencer-negative control siRNA were custom-made by RIBOBIO, China. cDNA clones for Gαtransducin (Gαt) or the carboxy-terminal domain of G protein-coupled receptor kinase 2 (GRK2ct) were described previously[35,36]. Mouse and human TRPA1 were gifts from Dr. Jordt (Duke University, USA). Mouse and human CXCR2 were purchased from OBiO (China). siRNA or plasmids were transfected into HEK293T cells using Lipofectamine™ 2000 according to the manufacturer's instructions. 4 h after the transfection, the medium was changed back to DMEM + 10% FBS, and the transfected cells were cultured for another 24–48 h before $Ca^{2+}$ imaging was launched

## Mouse DRG neuron culture

Mouse dorsal root ganglia (DRG) were dissected and dissociated with an enzyme solution containing 1 mg/ml collagenase A and 2 mg/ml dispase in DMEM for 1 h at 37 °C. After mechanical trituration and centrifugation, neurons were cultured in DMEM plus 10% FBS, 100 U/ml penicillin, and 100 µg/ml streptomycin on eight-well chambered coverglass (Nunc, Denmark) coated with poly-D-lysine (Sigma, USA) and mouse laminin (Thermo Fisher Scientific, USA).

## Whole-cell patch-clamp recording in DRG neurons

Whole-cell patch-clamp recordings were performed in primary cultured DRG neurons with Axopatch-200B amplifier and Digidata 1550B digitizer (Axon Instruments, USA)[44,65]. Patch pipettes with a resistance

of 3–5 MΩ were fabricated from hard borosilicate glasses using a pipette puller (P-97; Sutter Instruments, USA). Whole-cell recordings were performed with the extracellular solution containing (mM): NaCl 150, KCl 5, CaCl$_2$ 2.5, MgCl$_2$ 1, glucose 10, and HEPES 10 (pH 7.4 with NaOH). The internal pipette solution contained (in mM): KCl 140, MgCl$_2$ 1, CaCl$_2$ 0.5, EGTA 5, HEPES 10, and ATP 3 (pH 7.4 with KOH). The action potentials were evoked by a 500-ms inward current injection through the recording electrode under current clamp mode. Small diameter DRG neurons with Cm <42 pF were recorded as in previous studies[43,44,66]. Data were analyzed with Clampfit 10.2 (Axon Instruments).

## Ca$^{2+}$ imaging

Fura-2-based radiometric Ca$^{2+}$ imaging was performed as described previously[67]. For Ca$^{2+}$ imaging of HEK293 cells, cells were used within 48 h after transfection. For mouse DRG neurons, neurons were used 6 h after dissociation. The cells were loaded with 10 μM Fura-2 AM (Thermo Fisher Scientific, USA) in the loading buffer (containing 140 NaCl, 5 KCl, 2 CaCl$_2$, 2 MgCl$_2$, and 10 HEPES (in mM), pH 7.4 with NaOH) at 37 °C for 45 min in the dark. Cells were subsequently washed 3 times, rested for another 30 min in the dark, and then imaged. The cell images under the excitation wavelength of 340 and 380 nm were captured with a Nikon ECLIPSE Ti-S microscope (Nikon, Japan) with Polychrome V monochromator (Till Photonics, USA), Orca Flash 4.0 CCD camera (Hamamatsu, Japan) and MetaFluor software (Molecular Devices, USA). Fura-2 ratios ($F_{340}/F_{380}$) were used to reflect changes in intracellular Ca$^{2+}$ upon stimulation. Values were obtained from 20 to 60 cells in time-lapse images from each coverslip. A cell was considered responsive after the challenge if the peak Ca$^{2+}$ response was above 20% of the baseline as reported before[64].

## Western blot

Ankle joint samples and ipsilateral L3–L5 DRGs were harvested, then weighed and homogenized in RIPA buffer added with protease inhibitors. The supernatant was centrifuged at $15,000 \times g$ for 12 min at 4 °C. The protein concentrations of the lysates were determined by BCA assay. 20 μg of protein was loaded in each lane and separated on 8% SDS–PAGE gel, and then electrophoretically transferred to 0.45 μm polyvinyl difluoride membranes. The membranes were blocked with 5% non-fat milk in TBST solution at room temperature for 1 h and then incubated with primary antibodies at 4 °C overnight. The following primary antibodies were used: rabbit anti-CXCR2 (1:1000), rabbit anti-TRPA1 (1:500), and β-actin (1:5000) (Table S1). Subsequently, these blots were incubated with HRP-conjugated second antibodies (1:5000) for 2 h at room temperature (Table S1). The expression levels of targeted protein are normalized to the density of β-actin. The gel images were captured by FluoChem R (Biotechne, USA). Quantitative analysis was performed with ImageJ (NIH, USA). Unprocessed scans of the blots are provided in the Source Data file.

## Fluorescence-activated cell-sorting (FACS)

Ankle joints were collected, diced, and digested with 2 mg/ml dispase and 1 mg/ml collagenase type 1 (Thermo Fisher Scientific, USA) in RPMI 1640 + 10% fetal bovine serum (FBS) for 1 h at 37 °C. The cells were filtered through a cell strainer with a 70 μm nylon mesh (Corning, USA) and washed with RPMI 1640 + 10% FBS. Cells were then stained with a standard panel of immunophenotyping antibodies. A list of all antibodies used in the study is shown in Table S1, including anti-mouse CD11b (1:500) and anti-mouse Ly6G (1:500), for 1 h at room temperature. After staining, cells were suspended with a staining buffer; then, samples were captured using FACS Canto II Cytometry (BD Biosciences, USA), and the data were analyzed with FlowJo software (BD Biosciences, USA).

## In situ hybridization

In situ hybridization was performed using RNAscope® Multiplex Fluorescent Reagent Kit v2 (#322300-USM, Advanced Cell Diagnostics, USA) according to the manufacturer's instructions. The RNA probes (Mm-Trpa1-C1, #400211) were complementary to the target mRNAs. In each experiment, we used *Hs-ppib* (human peptidylprolyl isomerase B) and *Dapb* (bacterial dihydrodipicolinate reductase) probes as positive and negative controls, respectively. Fixed frozen samples (L3–L5 DRGs) were sliced with a frozen microtome (CryoStar NX50, Thermo Fisher) at 10 μm thickness. Pre-hybridization, hybridization, and washing were performed according to the manufacturer's instructions. Sections were then used for immunofluorescence staining to confirm the co-localization of *Trpa1* and CXCR2 in DRG neurons.

## Evans blue extravasation assay

To examine neurogenic inflammation in the hind paw, mice were anesthetized with 5% isoflurane. 100 μl of a 0.5% Evans blue (EB) solution was injected intravenously into anesthetized mice 10 min before neurogenic irritant application. AITC (5 mM, 10 μl) or CXCL5 (300 ng, 10 μl) were given by intraplantar injection. 3 h later, the mice were sacrificed. Plantar tissues were collected, weighted, and placed in 50% trichloroacetic acid solution, homogenized, centrifuged ($10,000 \times g$, 20 min), and the supernatant was mixed with ethanol at 1:3 ratios. EB content was quantified by measuring optical density at 620 nm using a microplate reader (SpectraMax M4, Molecular Devices, USA). Absorbance was normalized to per gram of tissue weight.

## Tissue homogenization and Elisa assay

Mice were terminally anesthetized and ankle joint or hind paw tissues were collected and diced. The diced tissues were homogenized using Bullet Blender (NextAdvance, USA) in 50 mM Tris-base (pH 7.4) and 150 mM NaCl with protease inhibitor (#04693132001, Roche, Switzerland) and 0.2% Triton-X. Homogenization was carried out for 20 min at full speed. The homogenates were centrifuged at $10,000 \times g$ for 10 min at 4 °C. The supernatant was tested by ELISA kit for the following analytes: CXCL5, SP, and CGRP (Table S1) according to the manufacturer's instructions. The plates were read at 450 and 570 nm using a microplate reader. Total protein was determined by BCA assay (Thermo Fisher Scientific, USA).

## Myeloperoxidase (MPO) activity measurement

Neutrophil recruitment in ankle joints was evaluated via quantification of the enzyme MPO activity using a commercial MPO detection kit (Elabscience, USA) as described previously[14]. Briefly, mice were terminally anesthetized, and the ankle joint was homogenized and centrifuged at $10,000 \times g$ at 4 °C for 15 min. 10 μl of the supernatant was transferred into PBS (pH 6.0) containing 0.17 mg/ml 3,3′,5,5′-tetramethylbenzidine and 0.0005% H$_2$O$_2$. MPO catalyzed the redox reaction of H$_2$O$_2$ and 3,3′,5,5′-tetramethylbenzidine and produced yellow-colored compounds, and the absorbance at 460 nm was determined. MPO activity was calculated and expressed as U/g tissue. One unit of MPO activity was defined as the quantity of enzyme that degraded 1 μmol H$_2$O$_2$ at 37 °C per g wet tissue.

## Histopathological assessment of ankle joint

Ankles were fixed with 10% paraformaldehyde in PBS and then decalcified for 21 days with EDTA and embedded in paraffin for further sectioning. The paraffin sections were stained with hematoxylin and eosin (H&E) for morphological analysis using a light microscope with ×10 and ×40 objectives. For histopathological analysis, the degrees of the following parameters were evaluated as reported[68]: (1) inflammatory infiltrate (from 0 = no inflammation, 1 = mild, 2 = moderate and 3 = severe inflammation); (2) cartilage injury (from 0 = no injury, 1 = mild, 2 = moderate and 3 = severe injury); (3) vascular proliferation (from 0 = no vascular proliferation, 1 = mild, 2 = moderate and

3 = severe vascular proliferation). The score was determined by adding up the three parameters for each sample. The analysis was performed in a blind manner using ×40 objectives. Three sections were obtained from each sample and then averaged to obtain the result.

## Multiplex assay of human serum samples

The stored serum samples were thawed and centrifuged at $10,000 \times g$ for 5 min at 4 °C to remove any debris. The Luminex Multiplex Assays (Luminex, USA), which were custom-designed, were used to determine the concentrations of a panel of inflammatory cytokines and chemokines according to the manufacturer's instructions. Briefly, analyte-specific antibodies are pre-encapsulated onto magnetic microparticles embedded in fluorophores, with a specific percentage of each particle region. The magnetic particles, standards, and samples are pipetted into the wells, and the immobilized antibody binds the target analyte. After washing away any unbound substances, a biotinylated antibody cocktail specific to the analytes of interest was added to each well. After the wash, streptavidin-phycoerythrin couples bound to biotinylated antibodies were added to each well. Following another wash, the microparticles were resuspended in a buffer and analyzed with the Luminex MAGPIX analyzer (Luminex, USA).

## Serum isolation and Elisa assay

Blood samples were collected from each mouse before and 3 h after rmCXCL5 injection and then processed by centrifugation at $1000 \times g$ for 15 min to isolate the serum. The concentration of CXCL5 in the serum samples was determined by Mouse CXCL5 Elisa Kit (Table S1) as recommended by the manufacturer. The plate was read at 450 nm using a microplate reader (Molecular Devices, USA).

## qPCR assay

Mouse ankle joint samples were collected and total RNA was extracted from tissues using an RNA isolation kit (TaKaRa Bio Inc, China) and reverse-transcribed to cDNA using Prime ScriptTM RT reagent Kit (TaKaRa Bio Inc, China). qPCR was performed with LightCycler 480 real-time PCR system (Roche, Switzerland) using LightCycler 480 SYBR Green I Master (Roche, Switzerland) with a 20 μl reaction system. The relative quantification of target gene expression was performed with the comparative cycle threshold method normalized to the level of a housekeeping gene β-actin. All reactions were performed in triplicate. The cycle threshold (CT) value was deduced from the analyzing software by LightCycler480 System. $^{\triangle\triangle}$CT method was used to calculate gene expression fold changes[69,70]. The sequences of the primers used are provided in Table S1.

## Co-immunoprecipitation (co-IP)

Ipsilateral L3–L5 DRGs from mice were dissected and then lysed in a buffer containing 50 mM Tris, 150 mM NaCl, 1 mM EDTA, and 0.2% Triton X-100 with protease inhibitor mixture. The procedures were conducted using an Immunoprecipitation Kit with Protein A + G Magnetic Beads (#P2179M, Beyotime, China) following the manufacturer's instructions. The lysates were centrifuged ($12,000 \times g$, 15 min) at 4 °C and 5% of the supernatant was taken for a whole-cell lysate sample. The remaining supernatant was precipitated with 1 μg of antibody at 4 °C overnight and, afterward, protein G beads at 4 °C for 2 h. The immunoprecipitated sample was denatured and prepared for immunoblotting.

## Generation of point mutations in TRPA1 channel

The mouse Trpa1 (NM_177781.5) was inserted into pCDNA3.1 vector (pTrpa1-WT) and mutation 1 (R1012A + R1017A), mutation 2 (R999A + K1000A + R1004A) and mutation 3 (R1012A + R1017A + R999A + K1000A + R1004A) plasmids were constructed by Repobio (Hangzhou, China). All the mutations were verified by gene sequencing.

## Computational modeling and simulation

The mouse TRPA1 (mTRPA1) structure was modeled using the SWISS-MODEL program based on the structure of human TRPA1 (PDB: 6V9W)[71], which has 80% sequence similarity. HDOCK is used for fast protein–protein docking[72]. The docking program first samples the putative binding modes between two proteins based global search method and then evaluates the sampled modes with a knowledge-based scoring function for protein–protein interactions. The model of mTRPA1 is used as the docking receptor, crystal structure of G protein beta gamma heterodimer (PDB: 6RMV)[42] is used as the ligand for docking. The binding mode of TRPA1 and G protein beta gamma heterodimer was determined based on the combination of TRPM3 and G protein complex structure and scoring function. Calculation of protein electrostatic surface potentials was performed by adaptive Poisson–Boltzmann solver (APBS)[73]. Visualization and analysis of model features were carried out by Open-Source Pymol (https://pymol.org).

## Statistical analysis

Data are presented as mean ± SEM. Statistical analyses were performed using GraphPad Prism 8.0 (GraphPad Software, USA). Normal distribution was assessed using the Shapiro–Wilk test, and homogeneity of variance was assessed by Levene's test. To compare between 2 groups, Student's *t*-test with two-tailed (parametric data) or Mann–Whitney *U* test (unpaired nonparametric data) was applied. To compare data among 3 or more groups, one- or two-way analysis of variance (ANOVA) followed by Bonferroni's post hoc test (parametric data) or Kruskal–Wallis test followed by Dunn's multiple comparisons (nonparametric data) was applied. The statistical details are documented in the figure legend. Statistical significance was accepted when $p < 0.05$.

## Reporting summary

Further information on research design is available in the Nature Portfolio Reporting Summary linked to this article.

# Data availability

RNA-Seq data has been deposited in NCBI GEO under accession code GSE242872. The structures referred to in this study are available in the Protein Data Bank under the accession codes PDB 6V9W and 6RMV. A reporting summary is available as a Supplementary Information file. The original data in this study are provided in the Source Data file. Source data are provided with this paper.

# Code availability

HDOCK is used for fast protein-protein docking (http://hdock.phys.hust.edu.cn/). Visualization and analysis of model features are carried out by Open-Source Pymol (https://pymol.org).

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

## Acknowledgements

This study was supported by the National Key R&D Program of China (2022YFC3500604 to Boyi L.), Zhejiang Provincial Natural Science Funds (LZ23H270001 to Boyi L.), National Natural Science Foundation of China (82305369 to Boyu L., 82105014 to Q.H., 81873365 to Boyi L.), Zhejiang Provincial Natural Science Funds (LQ21H270004 to Boyu L.) and research fund from Zhejiang Chinese Medical University (2021JKZDZC07 to Boyi L.). We thank Prof. Zhen-zhong Xu (Zhejiang University) for providing *Trpa1*⁻ᐟ⁻ and *Trpv1*⁻ᐟ⁻ mice, Prof. Naiming Zhou for providing Gα-transducin plasmid and Academy of Chinese Medical Sciences, Zhejiang Chinese Medical University for technical assistance.

## Author contributions

Boyi L. conceived the idea and supervised the project. C.Y., X.S., J.F., and C.W. designed the experiments. C.Y., Boyu L., Y.P., H.N., Y.Z., Q.H., and Y.T. performed animal experiments, immunostaining, Western blot, RNA scope, PCR, ELISA, and flow cytometry experiments. Z.D. performed electrophysiological recordings. C.P., X.W., and X.B. processed human serum and DRG samples. S.S. and H.A. performed structure modeling. C.Y., Boyu L., Z.D., and S.S. analyzed the data and prepared the figures. C.Y. and Boyi L. wrote the original draft of the manuscript.

## Competing interests

The authors declare no competing interests.
