## [Peer Review File · Nature Communications]

REVIEWER COMMENTS

Reviewer #1 (Remarks to the Author):

Thank you for this important piece of work that adds to the field of acute gout arthritis. Through a series of elegant experiments, the author report the role of novel neuronal CXCL5-CXCR2-TRPA1 axis in contributing to gout arthritis pain. In general the data support the conclusion. However , there are some clarifications and comments on the methods that are needed to support and strengthen the conclusions made regarding CXCL5 and its role in joint inflammation, particularly the demonstration of neutrophils and macrophages/ other inflammatory cytokines in the joints of mice model.

1. In the introduction, a stronger argument for the unmet need in acute gout treatment should be outlined, beyond the assumption that “constant use of these drugs results in severe side effects”. The key to gout management is in long-term control with urate argument for the unmet need in acute gout treatment could be strengthened by mentioning the limitations of NSAIDs/prednisolone in view of comorbidities that often accompany patients with gout eg chronic kidney disease and diabetes, limiting their frequent use

2. In reporting the results (eg pg 4, line 112) in mice model and ankle “edema” or inflammation, it is more accurate to state ankle diameter, a surrogate of ankle edema or swelling suggestive of inflammation.

3. In Figure 1, the author mentioned that “A monoclonal neutralizing antibody against CXCL5 or isotype control IgG was injected (3 µg/site) into ankle joint. The neutralizing antibody remarkably attenuated mechanical allodynia and ankle edema of model mice vs. control IgG (Fig. 1I-L)”. Did the authors check the change of neutrophil and macrophage levels at the ankle joint? This would be important to support the claim that CXCL5 contributes to joint inflammation in gout arthritis

4. Please clarify whether the experiments involved synovial fluid from the joint, synovium from ankle joint, or the ankle joint taken as a whole. It appears to be the latter

5. With reference to Figure 5C, please explain why CXCL5-induced mechanical allodynia was only able to be released around 50% by blocking CXCR2? Are there any other chemokine/receptor axis involved in this pain transduction signal?

6. Figure 6 L&M data showed CXCL5-induced mechanical allodynia and joint edema/ inflammation as measured by increase in diameter was markedly attenuated by CXCR2 deletion. Were other inflammatory cytokine levels in the ankle joint, like interleukins, TNF α and IFN measured? Likewise, this also applies to Figure 7.

7. Suggest to combine sections 5 and 6 (Figure 5, 6 and 7) and rename it to “CXCL5–induced mechanical allodynia and joint inflammation could be ameliorated by pharmacological blocking of CXCR2 or TRPA1”. The majority of details of the new mouse line with conditional *Cxcr2* deletion in peripheral nociceptive sensory neurons could go into the supplement, allowing the key highlights of results to be focused in the main text.

8. Figure 9 A & B data suggest that CXCR2/TRPA1 axis in DRG may not be the only reason for CXCL5-mediated allodynia as only around 50% of PWT and edema release could be rescued by CXCR2 selective knockout. What are the other potential working pathways? In Figure 1G & H, there were about 5-fold increase of CXCL5 protein expression after MSU induction, which could help to recruit neutrophils and macrophages to the ankle joint and evoke inflammation. In Figure 9J & K, neutrophils are markedly reduced by CGRP receptor antagonists BIBN (60 ng/site, i.pl.) or SP NK-1 receptor antagonist L-733060. Did you check macrophage count at the ankle joint with and without CGRP or SP inhibition?

9. In Figure 10, are you able to check CXCL5 levels in the synovial fluid from gout arthritis patients?

10. Fig 10 L-N: although the correlation between CXCL5 and IL6 ($R^2= 0.16$), serum uric acid ($R^2= 0.3$), IL-1 β (0.44) were statistically significant, these were not strong. Can you postulate why?

11. Please clarify whether CXCR1 expression is present in human DRG neurons.

Minor comments

12. In relation to Fig 2F, pls state in the results section (pg6 line 155) the % of CXCR2 co-expressed with CGRP+, IB4+ and NF 200+)

13. In Figure 3B, please provide a legend to denote the colours of the curves to aid interpretation of data. The colour legend from Fig 3C does not appear to apply

14. Pg 6 line 167, please clarify if you mean CXCR2-responding neurons and not CXCR5

15. Throughout the manuscript, kindly define the first time use of acronyms eg DRG etc. Likewise, in the footnotes of figures, all acronyms should be defined

16. Please check that all ref to figs in the results section is accurate eg Line 392: I believe the ref to the Fig is 10 L-N and not K-M

Reviewer #2 (Remarks to the Author):

This manuscript investigated the mechanisms underlying gout pain and inflammation and identified the axis CXCL5/CXCR2 as a major player involved in nociceptive sensory neuron hyperexcitability through TRPA1 (Transient Receptor Potential Ankyrin 1) activation. Neuronal CXCR2 deletion was shown to ameliorate joint pain and inflammation in a murine model of gout. Expression of CXCR2 and CXCL5 was confirmed in patients with gout arthritis.

This work is mostly confirmatory of previously published observations on the role of CXCR2 expression in dorsal root ganglia (DRG) in peripheral neuropathy. The selective ablation of CXCR2 obtained in SNS-Cre mice further supports the original statement, but the manuscript fails to improve our understanding of this phenomenon, such as clearly defining the biochemical basis of CXCR2 regulation of TRPA1 signaling; this is a still elusive and poorly defined issue. The proposed therapeutical values of AAV-PHP.s administration seems unrealistic. Results are often superficial in their nature, poorly described and overinterpreted; conclusions are not fully supported by in vitro and in vivo data.

Specific comments:

As a general comment, this study does not define the source of CXCL5 in gout joints. This is a crucial point that needs to be addressed. Similarly, histological evaluation of inflamed ankle joints is missing. The increase of diameter is not sufficient to accurately evaluate the extent of inflammation.

Figure 1D: in addition to DEGs simultaneously altered at 8 and 24hrs, a more detailed description of genes differentially expressed at the two time points could improve the comprehension of the results.

Figure 1F-H: the contribution of CXCL2, another ligand for CXCR2, should be investigated, in the gout pain. In addition, the potential role of IL-6 (the highest altered gene) in the pathogenesis of gout pain should also be investigated and discussed.

Figure 1I-L: the effect of anti-CXCL5 in neutrophils infiltrate in MSU ankle joint should be evaluated also by IHC.

Line 142: is CXCR2 expression in DRG modulated during MSU inflammation? in addition to IF data, a quantification of CXCR2 transcripts in DRG of untreated and MSU-treated mice should be included.

Figure 2B: a light micrographs showing the morphology and distribution of DRG in ankle joints should be provided.

Figure 2B-C: the IF images are of poor quality, a more detailed description (e.g., how many fields were evaluated?) should be included in figure legends. For instance, the statement that >95% of Neu are CXCR2 positive, does not appear to be supported by the enlarged merged image.

Figure 2D: a total black CXCR2^{-/-} picture is useless. Other control markers should be shown.

Figure 2F: light micrographs could help; same criticisms as Figure 2B-C. Line 157-160: the conclusions about the expression of CXCR2 in different neurons should be more detailed and complete.

Figure 3A: representative images do not help the comprehension of results. In Fig. 3B: legend should be improved with different coloured lines; quantitative graphs are also needed. How many cells were analysed? A statistical analysis is missing.

Line 162-170: the description is not accurate; a statistical analysis is missing.

Figure 3F: mCXCL1 showed a significant response when compared with the vehicle; its contribution should not be ignored.

Figure 3G: the same experiment should be performed also in the presence of mCXCL1 to clearly evaluate the functional contribution of the two chemotactic factors in the same experimental setting.

Figure 3N: a statistical analysis should be included.

Line 223: Figure 4F should be added to Fig A&B.

Line 226: Fig. 4G-J should be added.

Line 227: Fig. 4K-N should be instead of 4C-N.

Line 825, Figure 4D: the number of samples seems less than those indicated (no. 17).

Line 835, Figure 4O: the inclusion of a graph including bars and individual samples is suggested.

Line 243, Figure 5C: at 3, 6 and 9 hrs after injection, is the difference between the white and green bars, statistically significant? And how do the authors explain these differences?

Figure 6B: the addition of light micrographs is suggested.

Line 263: the authors refer to Fig. 6L&M and not Fig. 4.

Lines 263-265 As in Fig. 4L&M, CXCL5-induced mechanical allodynia and joint inflammation was markedly attenuated in SNS-Cre Cxcr2^{fl/fl} mice vs. SNS-Cre negative controls (Cxcr2^{fl/fl}). Where are these data shown?

Lines 266-269 Compared with WT controls, CXCL5-induced mechanical allodynia was significantly reduced in Trpa1^{-/-} but not in Trpv1^{-/-} mice (Fig. S4A). CXCL5-induced joint inflammation was significantly attenuated in Trpa1^{-/-} mice as well (Fig. S4B). In contrast to what authors said in the main

text Fig. S4A shows that mechanical allodynia was significantly reduced in *Trpv1*^{-/-} but not in mice *Trpa1*^{-/-}.

Line 268: the IHC analysis should be included.

Line 283: "Since we found CXCR2 largely co-expressed with CGRP+ DRG neurons by immunostaining (Fig. S1F)". Fig. S1F doesn't show immunostaining but a rmCXCL5 evoked Ca²⁺ responses (normalized with ionomycin response) in HEK293T cells transfected with different combinations of plasmids. The authors should refer to the correct figure.

Line 296, Fig. 7F: neutrophil infiltration in the inflamed joint should be evaluated by IHC.

Line 305: "reduced" could sound better than "ameliorate" referred to neutrophil infiltration.

Line 319: "CXCR2 immunostaining in ipsilateral L3-L5 DRG was stronger in model mice than control mice (Fig. 8B&C)". The difference in CXCR2 immunofluorescence shown in Fig. 8B does not support the quantification analysis (Fig. 8C).

Line 330, Fig. 7G: please, specify the time point of analysis after MSU injection.

Figure S6 could be moved as final figure and not just suppl.

Figure 9L: the representative images are not clear and useful and could be moved to supplementary material and leave only the quantification (Fig 9M).

Figure 10B: the IF images are of poor quality, a more detailed description of quantitative graphs (e.g., how many fields were evaluated?) should be included in figure legend.

Responses to the reviewers:

Reviewer #1

Thank you for this important piece of work that adds to the field of acute gout arthritis. Through a series of elegant experiments, the author report the role of novel neuronal CXCL5-CXCR2-TRPA1 axis in contributing to gout arthritis pain. In general the data support the conclusion. However , there are some clarifications and comments on the methods that are needed to support and strengthen the conclusions made regarding CXCL5 and its role in joint inflammation, particularly the demonstration of neutrophils and macrophages/ other inflammatory cytokines in the joints of mice model.

1. In the introduction, a stronger argument for the unmet need in acute gout treatment should be outlined, beyond the assumption that “constant use of these drugs results in severe side effects”. The key to gout management is in long-term control with urate argument for the unmet need in acute gout treatment could be strengthened by mentioning the limitations of NSAIDs/prednisolone in view of comorbidities that often accompany patients with gout eg chronic kidney disease and diabetes, limiting their frequent use

Answer: Thanks for the advice. We have added this important background information in the introduction, which helps to delineate the specific condition that limit the use of NSAIDs or prednisolone in gouty patients. The corresponding references have been added as well.

2. In reporting the results (eg pg 4, line 112) in mice model and ankle “edema” or inflammation, it is more accurate to state ankle diameter, a surrogate of ankle edema or swelling suggestive of inflammation.

Answer: We have changed ankle edema to ankle diameter according to the suggestion. Thanks.

3. In Figure 1, the author mentioned that “A monoclonal neutralizing antibody against CXCL5 or isotype control IgG was injected (3 µg/site) into ankle joint. The neutralizing antibody remarkably attenuated mechanical allodynia and ankle edema of model mice vs. control IgG (Fig. 1I-L)”. Did the authors check the change of neutrophil and macrophage levels at the ankle joint? This would be important to support the claim that CXCL5 contributes to joint inflammation in gout arthritis

Answer: In this revision, we performed some new experiments to answer this question. We performed IHC of the ankle joint and found that CXCL5 neutralizing antibody can significantly reduce the infiltrations of neutrophils and macrophages in ankle joint of gout model mice. This new data strengthens the claim that CXCL5 contributes to joint inflammation in gout arthritis. This data as well as the statistical summary are included as Suppl. Fig. 3A-D

4. Please clarify whether the experiments involved synovial fluid from the joint, synovium from ankle joint, or the ankle joint taken as a whole. It appears to be the latter

Answer: The whole ankle joint is collected and analyzed in our study. We have

emphasized this in methods section.

5. With reference to Figure 5C, please explain why CXCL5-induced mechanical allodynia was only able to be released around 50% by blocking CXCR2? Are there any other chemokine/receptor axis involved in this pain transduction signal?

Answer: Thanks for this question. We rechecked the data and we highly suspect this result could be due to insufficient pharmacological blockage of CXCR2 using SB225002 compound at lower dosage when time elapses. Because at 0.5 h time point, we observed near full recovery of the pain threshold vs. control mice. But when time elapses, the recovery gradually attenuates. In order to address this issue, we tried a higher dosage of SB compound (10 mg/kg, i.p.) in this revision. We found that 10 mg/kg SB can significantly alleviate mechanical pain vs. vehicle-treated mice, a result similar with 5 mg/kg dosage. Moreover, this higher dosage almost fully recovered the pain threshold vs. control mice (with no statistical difference between the two groups). This new panel has been put into Suppl. Fig. 4C. Thus, this result suggests that CXCL5-induced mechanical allodynia largely depends on CXCR2.

6. Figure 6 L&M data showed CXCL5-induced mechanical allodynia and joint edema/inflammation as measured by increase in diameter was markedly attenuated by CXCR2 deletion. Were other inflammatory cytokine levels in the ankle joint, like interleukins, TNF α and IFN measured? Likewise, this also applies to Figure 7.

Answer: Accordingly, we measured the gene expression of some typical inflammatory mediators, including *Il1b*, *Tnfa* and *Ifng*, in ankle joints by qPCR in this revision. Our new data showed that CXCL5 injection induced a remarkable increase in *Il1b*, *Tnfa* and *Ifng* gene expression in ankle joints of WT mice, whereas these increases induced by CXCL5 were significantly reduced in *Cxcr2* cKO mice (Fig. 6H-K). This result further helps to strengthen our conclusion.

7. Suggest to combine sections 5 and 6 (Figure 5, 6 and 7) and rename it to “CXCL5–induced mechanical allodynia and joint inflammation could be ameliorated by pharmacological blocking of CXCR2 or TRPA1”. The majority of details of the new mouse line with conditional *Cxcr2* deletion in peripheral nociceptive sensory neurons could go into the supplement, allowing the key highlights of results to be focused in the main text.

Answer: In this revision, we moved the original Fig. 5 and Fig. 6H-K to supplementary files accordingly. Since we are the first group to generate this conditional *Cxcr2* deletion in peripheral nociceptive sensory neurons, we think it will be important to show the validation of *Cxcr2* cKO in normal figures. Besides, we find it difficult to combine Fig. 6 with Fig. 7 into one single figure, since they contain so many panels and contents. Therefore, we still would like to keep Fig. 6 and Fig. 7 as separate figures. Please let us know if this is all right.

8. (1) Figure 9 A & B data suggest that CXCR2/TRPA1 axis in DRG may not be the only reason for CXCL5-mediated allodynia as only around 50% of PWT and edema

release could be rescued by CXCR2 selective knockout. What are the other potential working pathways?

Answer: Yes, indeed we observed that there is still some residual pain response in *Cxcr2* cKO mice. This could be due to complex nature of the inflamed ankle joint of gout model mice. During inflammation, multiple pro-inflammatory cytokines or chemokines are released to contribute to the overall pain and inflammation of gout arthritis. For example, in our recent study, we found that IL-33, a pro-inflammatory cytokine, is significantly increased in the inflamed ankle joint and contributes to gout arthritis pain by promoting neutrophil-dependent ROS production and TRPA1 channel activation through its receptor ST2 (PMID: 33204337). Our recent study also found that IL-1 β generated via NLRP3 inflammasome during gout arthritis contributes to gout arthritis pain by promoting TRPV1 channel overexpression in sensory neurons (PMID: 37464384). Besides, oxidative stress generated during gout arthritis can directly activate TRPA1 in sensory neurons to trigger pain in gout model mice. One recent study also highlights an important role of macrophage TRPV4's involvement in gout pain and inflammation (PMID: 34663597). Here we also observed *Il6* as the highest upregulated cytokine gene. IL-6 can promote functional expression of TRPV1 in DRG neurons and contributes to peripheral sensitization (PMID: 25775359). It is likely that IL-6 mediated signaling may also contribute to gout pain as suggested by reviewer#2. Therefore, CXCR2/TRPA1 signaling we identified in this study only constitutes one important aspect of gout arthritis pain mechanism. There are other potential signaling pathways documented to be involved in gout arthritis pain, reflecting the complexity of gout arthritis pain mechanisms. This explanation has been added in the discussion of this revised manuscript. Please check for more details. Thanks.

(2) In Figure 1G & H, there were about 5-fold increase of CXCL5 protein expression after MSU induction, which could help to recruit neutrophils and macrophages to the ankle joint and evoke inflammation. In Figure 9J & K, neutrophils are markedly reduced by CGRP receptor antagonists BIBN (60 ng/site, i.pl.) or SP NK-1 receptor antagonist L-733060. Did you check macrophage count at the ankle joint with and without CGRP or SP inhibition?

Answer: In this revision, we performed more experiments to check macrophages with CGRP or SP inhibition as suggested. We found that pharmacological blocking CGRP or SP receptor using BIBN or L-733060 significantly reduced macrophage infiltrations (indicated by Iba-1) in ankle joint tissues of gout model mice. This data has been added as Suppl. Fig. 9 in this revision.

9. In Figure 10, are you able to check CXCL5 levels in the synovial fluid from gout arthritis patients?

Answer: We were not able to collect synovial fluid from gouty patients at this moment, since this is not included in our current protocol. We are still planning to collect synovial fluids in our future experiments. But collecting synovial fluid is very difficult to achieve since most acute gout arthritis patients do not need synovial fluid withdrawal for examination or treatment. So it is difficult to collect the samples.

10. Fig 10 L-N: although the correlation between CXCL5 and IL6 ($R^2= 0.16$), serum uric acid ($R^2= 0.3$), IL-1beta (0.44) were statistically significant, these were not strong. Can you postulate why?

Answer: These data were obtained from gouty patients. As the reviewer knows, clinical data usually shows larger variations due to the differences in age, disease severity, physical condition, etc. Besides, we were only able to collect 37 samples. We believe a further increase in the number of patients may help to optimize these results. Here we deleted IL-6 panel since the correlation is not that high.

11. Please clarify whether CXCR1 expression is present in human DRG neurons.

Answer: We tested *Cxcr1* gene expression from human DRG cDNA by qPCR and barely found any *Cxcr1* gene expression (Fig. 10A). Furthermore, we analyzed published single-cell sequencing dataset from human DRG (GSE201586, GSE169301. PMID: 36690629, 35503034) and found that *Cxcr1* gene showed minimal expression in human DRG neurons. Considering the minimal expression of *Cxcr1* gene in human DRG, we here postulate that CXCR1 is unlikely to be highly present in human DRG neurons. Immunostaining of human DRG using commercial CXCR1 antibody may be another option. However, we are in lack of a positive control to test the antibody's validity in human tissues.

Minor comments

12. In relation to Fig 2F, pls state in the results section (pg6 line 155) the % of CXCR2 co-expressed with CGRP+, IB4+ and NF 200+).

Answer: Done. Please see result part for details. Moreover, we modified the statistics in Fig. 2F to pie chart, which is easier to discern the co-expression rate.

13. In Figure 3B, please provide a legend to denote the colors of the curves to aid interpretation of data. The color legend from Fig 3C does not appear to apply

Answer: This is done. The colors denote corresponding cells recorded in panel A. Please see Fig. 3 legend for details.

14. Pg 6 line 167, please clarify if you mean CXCR2-responding neurons and not CXCR5.

Answer: It should be CXCL5 but not CXCR5. This error has been modified. Sorry.

15. Throughout the manuscript, kindly define the first time use of acronyms eg DRG etc. Likewise, in the footnotes of figures, all acronyms should be defined.

Answer: Done.

16. Please check that all ref to figs in the results section is accurate eg Line 392: I believe the ref to the Fig is 10 L-N and not K-M.

Answer: Done. Sorry for the error. Thanks.

Reviewer #2

This manuscript investigated the mechanisms underlying gout pain and inflammation and identified the axis CXCL5/CXCR2 as a major player involved in nociceptive sensory neuron hyperexcitability through TRPA1 (Transient Receptor Potential Ankyrin 1) activation. Neuronal CXCR2 deletion was shown to ameliorate joint pain and inflammation in a murine model of gout. Expression of CXCR2 and CXCL5 was confirmed in patients with gout arthritis.

This work is mostly confirmatory of previously published observations on the role of CXCR2 expression in dorsal root ganglia (DRG) in peripheral neuropathy. The selective ablation of CXCR2 obtained in SNS-Cre mice further supports the original statement, but the manuscript fails to improve our understanding of this phenomenon, such as clearly defining the biochemical basis of CXCR2 regulation of TRPA1 signaling; this is a still elusive and poorly defined issue. The proposed therapeutical values of AAV-PHP.s administration seems unrealistic. Results are often superficial in their nature, poorly described and overinterpreted; conclusions are not fully supported by in vitro and in vivo data.

Answer: As the reviewer knows, since Wilson et al. from Bautista's group found that activation of GPCR MrgprA1 by chloroquine can trigger TRPA1 activation, several studies pop up showing that certain GPCR activation can lead to TRPA1 activation, including TSLPR, HTR7, S1PR3, TGR5, etc. However, no detailed delineations of the exact biochemical basis underlying CXCR2 regulation of TRPA1 have been published ever since. Thus, this remains to be a challenging question in TRPA1 study field. According to the reviewer's suggestion, we attempted to solve this challenging question. To address this issue, we made good use of the HEK293T cell expression system. Our initial results indicate that mCXCR2 can form functional interaction with mTRPA1 that can trigger mTRPA1 activation upon mCXCL5 application (Fig. 4A-F). Importantly, previous work from Bautista and Bunnet's groups suggests that G $\beta\gamma$ could serve as a downstream signaling of GPCR to activate TRPA1 via pharmacological methods. Therefore, we put special focus on G $\beta\gamma$ in our following study.

We performed several lines of experiments to explore the contribution of G $\beta\gamma$ to TRPA1 activation by GPCR. We tested whether the sequestration of endogenous G $\beta\gamma$ subunits can block mTRPA1 activation by mCXCR2. Transfecting HEK293T cells with either G α transducin (G α t) or the carboxy-terminal domain of G protein-coupled receptor kinase 2 (GRK2ct), which are both well-known scavengers of G $\beta\gamma$ subunits (PMID: 9596582, 12858180), significantly reduced mTRPA1 activation by mCXCR2 upon CXCL5 challenge (Fig. 4G). We further used siRNA technique to knockdown the most abundantly expressed G β isoforms, G β 1 and G β 2, in HEK293T cells (PMID: 36805843). *Gnb1* and *Gnb2* gene knockdown significantly reduced mTRPA1 activation by mCXCR2 upon CXCL5 challenge (Fig. 4G). It is known that AITC activates TRPA1 via direct covalent modification of cysteine residuals in TRPA1 channel (PMID: 17164327). We found that neither sequestration of endogenous G $\beta\gamma$ or knockdown of G β 1 and G β 2 expression has any significant effect on AITC-induced TRPA1 activation (Fig. 4H). These results clearly indicates that CXCR2 triggers TRPA1 activation via

G $\beta\gamma$ signaling.

After the dissociation from G α upon GPCR activation, G $\beta\gamma$ can modulate ion channel activity via direct binding to the channel or via triggering downstream signaling pathways (PMID: 33085809, 29776603). Therefore, we tested if G $\beta\gamma$ -related downstream signaling pathways contribute to TRPA1 activation. Pharmacological blocking G $\beta\gamma$ -related downstream signaling, including PLC, PI3K, PKA and MAPK has no significant effect on mTRPA1 activation by mCXCR2 upon CXCL5 challenge. Similarly, blocking these pathways has no effect on AITC-induced TRPA1 activation (Fig. 4G). To further explore if G $\beta\gamma$ can directly bind with TRPA1 upon CXCR2 activation, we performed co-IP assays in HEK293T cells. As shown in Fig. 4I&J, co-IP revealed stronger coupling of G β with TRPA1 upon CXCR2 activation by CXCL5. These results indicate that G $\beta\gamma$ can bind with TRPA1 upon CXCR2 activation.

To further explore the structure basis of CXCR2 regulation of TRPA1, we collaborated with structure biologists to examine the possible binding sites of G $\beta\gamma$ with mouse TRPA1. We performed computer-aided simulation and structure-based modeling to predict mTRPA1 and G $\beta\gamma$ binding model. The docking model shows that G β binds to the intracellular structural domain of mTRPA1, with K992-M1018 in mTRPA1 being the region of direct interaction, and the secondary structure of this region shows a helix and β -folding (Fig.4K-N). The region we predicted where mTRPA1 binds to G β is located at the C-terminal end of the TRP helix, and G β binds to TRPA1 at the same location as it does to TRPM3 (PMID: 33122432). Electrostatic interaction analysis showed that the K992-M1018 segment of mTRPA1 exhibits a high potential feature, which perfectly complements the low potential of the G β -interacting region (Fig. 4N). As shown in Fig. 4O, there are 5 key residues in the electrostatic interaction of mTRPA1 with G β , including R1012/R1017 in region A and R999/K1000/R1004 in region B. Similar to a reported working model for TRPM3-G $\beta\gamma$ complex, K992-M1018 of TRPA1 does not interact with spatially distant G γ (PMID: 33122432). We then performing point mutations of these key residuals to see if they can affect G β binding with mTRPA1 and subsequently affect mTRPA1 activation by mCXCR2. Mutation of the 2 residues (R1012 &R1017) located in region A only slightly reduced TRPA1 activation. Mutation of the 3 residues (R999, K1000&R1004) located in region B significantly reduced mTRPA1 activation induced by mCXCR2. Furthermore, mutations of all 5 residues caused further reduction of mTRPA1 activation by mCXCR2 (Fig. 4P).

In addition, we have ascertained beforehand that all channel proteins carrying these point mutations were still functional, since AITC evoked similar responses among wildtype and all these mutated channels (Fig. 4P). Finally, we tested if the mutated TRPA1 channel exhibits reduced binding with G β by co-IP experiments. As shown in Fig. 4Q, the binding of TRPA1 channel with G β was markedly reduced by mutations of all 5 residues. Therefore, we have identified the key regions as well as the residues located in TRPA1 that are involved in binding with G β , which results in channel activation. Based upon these results, we speculate that the binding of G β to the K992-M1018 segment of TRPA1 causes the outward movement of the TRP helix, which in

turn pulls TM6 to undergo metastasis and thus opens the lower gate of the TRPA1 channel (Fig. 4R). But it should be noted that CXCR2-induced TRPA1 activation was not completely eliminated by these mutations we identified. It remains likely that some other regions may also contribute to G β -binding to TRPA1 and subsequent channel activation. Moreover, it should also be noted that protein-protein docking cannot fully map the full details of protein interactions, therefore the microscopic mechanisms of protein interactions will need to be further addressed via cryo-electron microscopy. Please see new Fig. 4 and results and discussion part for more details.

In all, we have performed several lines of experiments to systematically explore the biochemical basis of CXCR2 regulation of TRPA1 activation as requested in this revision. These experiments took us nearly 6 months to finish and we have to request an extension of our submission from the editor. But these new results have significantly improved the quality of our work and help to solve a long-time unanswered question in TRPA1 field. We thank the reviewer for this constructive suggestion. According to the reviewer, we also deleted the data of AAV-PHP.s to make room for other important experiments.

Specific comments:

1. As a general comment, this study does not define the source of CXCL5 in gout joints. This is a crucial point that needs to be addressed. Similarly, histological evaluation of inflamed ankle joints is missing. The increase of diameter is not sufficient to accurately evaluate the extent of inflammation.

Answer: In this revision, we explored the cellular source of CXCL5 in ankle joints of gout model mice. We performed IHC using markers for different types of cells. According to literatures, CXCL5 can be produced by macrophages, mast cells and fibroblasts, etc. and these cells can all be activated by MSU. Therefore, we used markers for these specific types to explore the inflamed ankle joint of gout model mice. Immunostaining showed that a majority (56.6%) of CXCL5 positively stained cells co-stained with fibroblast cell marker vimentin. To a less extent, 30% and 12.3% of CXCL5 positively stained cells co-stained with markers for macrophage (Iba-1) and mast cell (avidin), respectively. Therefore, this result suggests that CXCL5 can be produced from several types of cells in the inflamed joint, including fibroblasts, macrophages and mast cells, under gout arthritis condition. This result has been added as Suppl. Fig. 2. In this revision, we also performed more experiments to provide IHC staining to study inflammatory cell infiltration (e.g. Suppl. Fig. 3, 7&9) as well as qPCR to study inflammatory cytokine expression (e.g. Fig. 6H-K, Fig. 9L), which is also suggested by Reviewer#1. We believe these methods together can help to accurately evaluate the inflammation.

2. Figure 1D: in addition to DEGs simultaneously altered at 8 and 24hrs, a more detailed description of genes differentially expressed at the two time points could improve the

comprehension of the results.

Answer: We have included two KEGGs to analyze the signaling pathways of these DEGs potentially involved at 8 and 24 h time points as supplementary files (Suppl. Fig. 1) in this revised manuscript.

2. Figure 1F-H: the contribution of CXCL2, another ligand for CXCR2, should be investigated, in the gout pain. In addition, the potential role of IL-6 (the highest altered gene) in the pathogenesis of gout pain should also be investigated and discussed.

Answer: In this revision, we evaluated the contribution of CXCL2 to gout pain and inflammation accordingly. We found that blocking CXCL2 by applying neutralizing antibody to the ankle joints could not achieve similar analgesic effect as CXCL5 antibody (Fig. 1I). But it showed some beneficial effects on joint inflammation (Fig. 1J). IL-6 is indeed the highest altered gene in our result. We are now in the process of testing IL-6's involvement in gout arthritis pain. But this piece of work falls out of the scope of present study, which focuses on chemokine signaling CXCL5/CXCR2's involvement in gout pain. Instead, we added a paragraph to discuss the potential involvements of IL-6 (e. g. its potentiating effect on TRPV1) and several other potential cytokines and signaling pathway's involvement in gout arthritis pain, which is also suggested by Reviewer#1. Hence, we would like to report IL-6's effect as a separate study if everything goes well in our future explorations. Thanks.

3. Figure 1I-L: the effect of anti-CXCL5 in neutrophils infiltrate in MSU ankle joint should be evaluated also by IHC.

Answer: We performed new IHC experiments to explore the effect of anti-CXCL5 on neutrophils and macrophages as suggested. We found that blocking CXCL5 significantly reduced the infiltrations of neutrophils and macrophages in periarticular tissues of ankle joint of gout model mice. This data has been put into Suppl. Fig. 3A-B.

4. Line 142: is CXCR2 expression in DRG modulated during MSU inflammation? In addition to IF data, a quantification of CXCR2 transcripts in DRG of untreated and MSU-treated mice should be included.

Answer: We reported that CXCR2 expression is significantly up-regulated in DRG neurons during MSU-induced inflammation as in Fig. 8. We used both Western blot and IHC for the quantification of CXCR2 protein in DRG (Fig. 8A-C). In addition, Ca^{2+} imaging is used for functional assay (Fig. 8F&G). Co-IP is used for confirmation (Fig. 8D&E). Therefore, we do not think adding qPCR is very much needed for the quantification as said in this case. Please let us know if our answers are all right. If not, we can still prepare to perform qPCR as requested.

5. Figure 2B: a light micrographs showing the morphology and distribution of DRG in ankle joints should be provided.

Answer: During our initial IHC imaging, we did not capture pictures of light micrograph since people usually do not show light micrograph in their immunostaining. In IHC, the distribution of DRG can be specifically outlined by NeuN staining, whereas

the overall outline and all cells can be identified by DAPI staining, as already included in our IHC. Without the aid of light micrograph does not hamper the interpretation of our results. But we do feel sorry that we could not recapture these light pictures because these sections were prepared and analyzed more than two years ago. Redoing the staining for the purpose of adding light micrographs will cause too much extra time and resources. We hope the reviewer can understand our situation. We will plan to capture light micrographs in all of our future IHC staining, which is a very good advice for us. Thanks.

6. Figure 2B-C: the IF images are of poor quality, a more detailed description (e.g., how many fields were evaluated?) should be included in figure legends. For instance, the statement that >95% of Neu are CXCR2 positive, does not appear to be supported by the enlarged merged image.

Answer: We feel sorry that during our initial submission, we only provided low quality images that has been compressed to reduce the file size since the journal has strict file size limits. So the pictures may look in not good quality during initial submission. In this revision, we have provided higher quality images. We hope this time the reviewer can find it much improved. We evaluated 3 fields per mouse and then make an average. We evaluate 4-5 mice per group in total as indicated. This info has been added in the figure legends. Our original statement is that “CXCR2 was expressed in almost all WGA-labeled ankle-joint innervating DRG neuron (>95%).” This means that >95% of WGA positive cells are CXCR2 positive. This is actually well reflected by the representative pictures as shown. But please be aware that this result is calculated from the overview pictures of IHC staining (as shown in upper panels), but not deduced from the enlarged pictures. As can be seen, the enlarged pictures only include around 3 WGA positive cells, which is hard to perfectly correlate with the results we obtained from the overviewed pictures. Thanks.

7. Figure 2D: a total black CXCR2^{-/-} picture is useless. Other control markers should be shown.

Answer: NeuN staining has been added as the background staining in the revised figure (Fig. 2D). Now it clearly shows that CXCR2 global KO mice showed no CXCR2 staining in the DRG.

8. Figure 2F: light micrographs could help; same criticisms as Figure 2B-C. Line 157-160: the conclusions about the expression of CXCR2 in different neurons should be more detailed and complete.

Answer: Please refer to our answer to Q#5. We will remember to capture light micrographs in all of our future studies when doing IHC. Thanks for the advice. Now the image quality has been improved after we upload larger files. We hope the reviewer can see it with better details. We have provided more detailed descriptions of the overlaying percentages in the results part. Pie charts are used instead for more clarity. Please see for more details.

9. Figure 3A: representative images do not help the comprehension of results. In Fig. 3B: legend should be improved with different colored lines; quantitative graphs are also needed. How many cells were analyzed? A statistical analysis is missing.

Answer: The representative images in Fig. 3A denote the cells responding to different challenges. The arrows indicated the cells showing responses to CXCL5. The responses of these arrow-indicated cells were further illustrated in Fig. 3B. We now used different colored lines to show these 5 specific cells recorded in Fig. 3A. The quantification and statistical analysis of the percentage of responding cells is further shown in Fig. 3F. Each Ca²⁺ imaging recording usually include 40-60 neurons. Therefore, for a result that contains 6 data points, it will usually include around 240-360 neurons for the calculation. This info has been added in the legends.

10. Line 162-170: the description is not accurate; a statistical analysis is missing.

Answer: We have improved our descriptions with specific statistics added in these lines. The Venn diagram is used for showing the overlapping with AITC/Capsaicin in Fig. 3C, whereas the total number of neurons calculated is illustrated (493 and 486, respectively). The statistical analysis of % responding neurons to CXCL5 is further displayed in Fig. 3F.

11. Figure 3F: mCXCL1 showed a significant response when compared with the vehicle; its contribution should not be ignored.

Answer: Yes, CXCL1 also triggers Ca²⁺ response, although to a much less extent vs. CXCL5. We tested if blocking CXCL1 using its neutralizing antibody could affect gout pain and inflammation in this revision as suggested. We found that blocking CXCL1 cannot produce significant alleviation of gout pain compared with CXCL5 blockage (Fig.11). We guess this could be due to two facts: on one hand, CXCL1 cannot produce strong Ca²⁺ response in DRG neurons as compared with CXCL5 at same dosage. On the other hand, CXCL1 gene upregulation in the inflamed joint of gout model mice is much smaller than CXCL5. At 8 and 24 h time points, the fold change of Cxcl5 gene is 16.79 and 3.56, whereas for Cxcl1 gene, it is 3.34 and 1.32, respectively. Therefore, these results indicate that the overall contribution of CXCL1 to gout arthritis pain is not comparable to that of CXCL5.

12. Figure 3G: the same experiment should be performed also in the presence of mCXCL1 to clearly evaluate the functional contribution of the two chemotactic factors in the same experimental setting.

Answer: Accordingly, we did these experiments with mCXCL1. The new data showed that mCXCL1-induced Ca²⁺ response can be attenuated by denature of the protein and by blocking CXCR2 and in neurons derived from Cxcr2 KO mice, all of which is similar with mCXCL5. Please see new Fig. 3G (right panel) for details.

13. Figure 3N: a statistical analysis should be included.

Answer: The statistical analysis is shown on the right panel, showing 44.2% of CXCR2⁺ cells expressing TRPA1, which is derived from a total of 267 CXCR2⁺ cells.

This result is obtained from 6 observational fields derived from 3 mice. This info has been added in the figure legend.

14. Line 223: Figure 4F should be added to Fig A&B. Line 226: Fig. 4G-J should be added. Line 227: Fig. 4K-N should be instead of 4C-N.

Answer: Thanks. These sentences have been modified accordingly.

15. Line 825, Figure 4D: the number of samples seems less than those indicated (no. 17).

Answer: Actually, in our initial submission, some data points were overlapped with each other, so that it seemed less than as indicated. Since AP numbers are all whole numbers, they are much easier to be the same with others and overlap. In this revision, we shifted some of the overlapped data points either to the right or left, so that they are visible now. We also double checked the numbers in each panel and corrected some errors. The exact value and numbers can be found in the Source File accompanying this submission.

16. Line 835, Figure 4O: the inclusion of a graph including bars and individual samples is suggested.

Answer: Done. It is now Fig. 5O.

17. Line 243, Figure 5C: at 3, 6 and 9 hrs after injection, is the difference between the white and green bars, statistically significant? And how do the authors explain these differences?

Answer: Reviewer #1 also raised the same question. We rechecked the data and we highly suspect this result could be due to insufficient pharmacological blockage of CXCR2 using SB225002 compound at lower dosage (5 mg/kg) when time elapses. Because at 0.5 h time point, we observed near full recovery of the pain threshold vs. control mice. But when time elapses, the recovery gradually attenuates. In order to address this issue, we tried a higher dosage of SB compound (10 mg/kg, i.p.) in this revision. We found that 10 mg/kg SB can significantly alleviate mechanical pain vs. vehicle-treated mice, a result similar with 5 mg/kg dosage. Moreover, this higher dosage almost fully recovered the pain threshold vs. control mice (with no statistical difference between the two groups) when time elapses. This new panel has been put into Suppl. Fig. 4C. Thus, this result suggests that CXCL5-induced mechanical allodynia largely depends on CXCR2.

18. Figure 6B: the addition of light micrographs is suggested.

Answer: Please refer to our response to Q#5. Thanks.

19. Line 263: the authors refer to Fig. 6L&M and not Fig. 4.

Answer: Sorry. It has been changed.

20. Lines 263-265 As in Fig. 4L&M, CXCL5-induced mechanical allodynia and joint

inflammation was markedly attenuated in SNS-Cre *Cxcr2*^{fl/fl} mice vs. SNS-Cre negative controls (*Cxcr2*^{fl/fl}). Where are these data shown?

Answer: These data are now illustrated in Fig. 6F-G.

21. Lines 266-269 Compared with WT controls, CXCL5-induced mechanical allodynia was significantly reduced in *Trpa1*^{-/-} but not in *Trpv1*^{-/-} mice (Fig. S4A). CXCL5-induced joint inflammation was significantly attenuated in *Trpa1*^{-/-} mice as well (Fig. S4B). In contrast to what authors said in the main text Fig. S4A shows that mechanical allodynia was significantly reduced in *Trpv1*^{-/-} but not in mice *Trpa1*^{-/-}.

Answer: We actually mean CXCL5-induced mechanical allodynia was significantly “ameliorated” in *Trpa1*^{-/-} but not in *Trpv1*^{-/-} mice. The word “REDUCED” has now been changed to “ameliorated” to avoid any misunderstanding in the result part. Please let us know if this works. Thanks.

22. Line 268: the IHC analysis should be included.

Answer: We have performed new IHC staining and analysis of neutrophils in *Trpa1* KO mice. The results indicate that CXCL5-induced neutrophil infiltration is significantly reduced in *Trpa1* KO mice. Please see Suppl. Fig. 7C-D for details.

23. Line 283: “Since we found CXCR2 largely co-expressed with CGRP+ DRG neurons by immunostaining (Fig. S1F)”. Fig. S1F doesn’t shows immunostaining but a rmCXCL5 evoked Ca²⁺ responses (normalized with ionomycin response) in HEK293T cells transfected with different combinations of plasmids. The authors should refer to the correct figure.

Answer: Sorry for this error. It should refer to Fig. 2F instead. We have changed it accordingly.

24. Line 296, Fig. 7F: neutrophil infiltration in the inflamed joint should be evaluated by IHC.

Answer: Thanks for the advice. We performed new IHC staining of neutrophils to compare WT vs. *Cxcr2* cKO mice. The results indicate that neutrophil infiltration induced by CXCL5 is significantly reduced in *Cxcr2* cKO mice (Suppl. Fig. 7E-F), a result consistent with our MPO result in Fig. 7F.

25. Line 305: “reduced” could sound better than “ameliorate” referred to neutrophil infiltration.

Answer: The word “ameliorate” has replaced “reduced”. Thanks for the suggestion.

26. Line 319: “CXCR2 immunostaining in ipsilateral L3-L5 DRG was stronger in model mice than control mice (Fig. 8B&C)”. The difference in CXCR2 immunofluorescence shown in Fig. 8B does not support the quantification analysis (Fig. 8 C).

Answer: Sorry, this misconception could be due to the low image quality in the first round of submission. In this revision, a higher quality images have been uploaded

instead. As can be clearly seen in Fig. 8B&C, CXCR2 immunostaining intensity is significantly up-regulated, since more cells showed higher CXCR2⁺ fluorescence in model mice than control mice. Besides, this finding is further supported by Western blot and Ca²⁺ imaging results. Thus, these results in together support that CXCR2 expression is functionally upregulated in DRG neurons of gout model mice.

27. Line 330, Fig. 7G: please, specify the time point of analysis after MSU injection.

Answer: The time point for analysis is 8 h after MSU injection.

28. Figure S6 could be moved as final figure and not just suppl.

Answer: The table has been moved to final figure as Fig. 9L, and thanks for the suggestion.

29. Figure 9L: the representative images are not clear and useful and could be moved to supplementary material and leave only the quantification (Fig 9M).

Answer: We deleted the representative images of gait. The quantification analysis is left unchanged and laid along the bottom of the figure.

30. Figure 10B: the IF images are of poor quality, a more detailed description of quantitative graphs (e.g., how many fields were evaluated?) should be included in figure legend.

Answer: Again, the low quality is due to the initial submission of compressed files. This time we have provided the full size high quality images that are much better than before. 6 fields were calculated. This info has been added to the figure legend.

REVIEWER COMMENTS

Reviewer #1 (Remarks to the Author):

Thank you for the greatly revised manuscript delineating the the biochemical basis of CXCR2 regulation of TRPA1 signaling. CXCR2 was shown to be the key mediator of CXCL5 in nociceptive sensory neurons.

In the revised Fig 9E, substantial neutrophil infiltration is still seen in the joints of the CXCR2 knockout mice. What might be the reason for this? e.g. could this neutrophil infiltration be working through neutrophils' other receptor - CXCR1? Although your data showed that the CXCR1 expression in DRG is undetectable, it does express on both neutrophils and monocytes. This recruitment of neutrophils and macrophages to the inflamed joints could mediate through CXCL5-CXCR1 axis when CXCR2 is blocked.

For Sup Fig. 3A & C staining, I suggest you do the multiple fluorescent IHC staining with DAPI (for cellular nucleus), CD45 (for leukocytes) and ly6G (for neutrophils) to exclude the non-specific signals. These non-specific signals were observed from your single antibody staining. Beyond that, a basic H&E staining is necessitated to define the joint structure. H&E and multiple fluorescent IHC staining protocols are applied to all other imaging analysis in this manuscript.

All other prior clarifications have been addressed

Reviewer #3 (Remarks to the Author):

The manuscript by Liu and co-authors showing us a very interesting and important work. CXCL5/neuronal CXCR2/TRPA1 axis contributes to gout arthritis pain, neutrophil influx and joint inflammation. Here, the study corroborates the CXCL5 overexpressed in animal models and serum from gout arthritis patients. Neuronal CXCR2 deletion was shown to ameliorate joint pain and inflammation in the animal model of gout arthritis. Moreover, neuronal CXCR2 or TRPA1 deletion prevented the hyperexcitability of DRG nociceptive sensory neurons induced by the CXCL5 administration. This study further revealed the neuronal CXCR2 coordinates with neutrophil CXCR2 to contribute to CXCL5-induced neutrophil chemotaxis by promoting CGRP and SP release and plasma extravasation in animals with gout arthritis. Additionally, this work first confirmed signaling of G β γ 's (CXCR2 signaling) contribution to activate TRPA1 by showing that G β γ sequestration or G β knockdown reduced CXCR2-mediated TRPA1 activation.

Overall, Liu's work provide valuable evidence that CXCL5/CXCR2/TRPA1 signaling as a major player involved in gout arthritis pain. It will help us to further expand the knowledge of nociceptive sensory neurons-immune interactions in pain and inflammation. The manuscript is clearly written and easily readable. In my opinion the revised manuscript has well addressed the reviewer #2's comments. However, I have several minor concerns about the manuscript.

1. Fig. 5, the analyses of the electrophysiology data are solid but lack detail. More electrophysiological parameters for the DRG neurons (e.g. RMP, Cm, rheobase current...) should be presented in a table.
2. Line 742, AP were evoked by a 600-ms step current, while in the Fig 5C, the schematic diagram marked as 500-ms of step current.
3. Line 773-774, is there any evidence or literature references to support the Cm < 42 pF is a small diameter DRG neuron?
4. The present results shown that the CXCL1 or CXCL2 mediate inflammation (Fig1I&J) via the receptor CXCR2, and CXCL1 also produced Ca²⁺ response in mouse DRG neurons (Fig 3F). These points should be addressed in the discussion.

Reviewer #1 Comments:

Thank you for the greatly revised manuscript delineating the biochemical basis of CXCR2 regulation of TRPA1 signaling. CXCR2 was shown to be the key mediator of CXCL5 in nociceptive sensory neurons.

1. In the revised Fig 9E, substantial neutrophil infiltration is still seen in the joints of the CXCR2 knockout mice. What might be the reason for this? e.g. could this neutrophil infiltration be working through neutrophils' other receptor - CXCR1? Although your data showed that the CXCR1 expression in DRG is undetectable, it does express on both neutrophils and monocytes. This recruitment of neutrophils and macrophages to the inflamed joints could mediate through CXCL5-CXCR1 axis when CXCR2 is blocked.

Answer: We first thank the reviewer for the positive comments for our revised manuscript. As the reviewer pointed out, indeed there was still certain neutrophil infiltration remained in ankle joints of CXCR2 cKO mice as seen in Fig. 9E. But it should be noted that this data was derived from mice that specifically lacked *Cxcr2* gene expression in nociceptive sensory neurons, whereas *Cxcr2* gene expression on neutrophils still remained intact (Fig. 6B and Suppl. Fig. 5A-D). Thus, neutrophil influx can still be initiated through CXCL5-neutrophilic CXCR2 axis (although to a significantly less extent without the enhancing effect from neuronal CXCR2). Therefore, the remaining neutrophil influx observed could be due to neutrophilic CXCR2-mediated effect.

In our study, we found that *Cxcr2* cKO mice showed significantly attenuated neutrophil influx (Fig. 9C-G). In combination with both in vivo and in vitro results obtained from Fig. 7 and Fig. 9, these series of experiments unraveled a critical contribution of neuronal CXCR2 to enhancing neutrophil chemotaxis in gout arthritis via triggering CGRP and substance P-mediated vasodilation and plasma extravasation (graphical abstract in Suppl. Fig. 11). Therefore, this finding reveals an important function of neuronal CXCR2 in immune modulation that has never been identified before. However, it should be noted that CXCR2 is not the sole participant involved in neutrophil chemotaxis during gout arthritis. As suggested by the reviewer, it also remains likely that CXCR1 may mediate the residual neutrophil infiltration observed. In addition, there could still be other important mechanisms contributing to neutrophil chemotaxis as well. For example, in our recent study, we found that IL-33 was significantly increased in the inflamed ankle joint of gout model mice and contributes to neutrophil infiltration (Yin et al., *Theranostics*, 2020, PMID: 33204337). IL-33 acts on ST2 receptor expressed on neutrophils to directly recruit neutrophils and serves as a potent neutrophil chemoattractant especially under inflammation condition (Verri et al., *Ann Rheum Dis*. 2010, PMID: 20472598). Therefore, the residual neutrophil influx observed in *Cxcr2* cKO mice could possibly due to multiple reasons as mentioned above.

2. For Sup Fig. 3A & C staining, I suggest you do the multiple fluorescent IHC staining with DAPI (for cellular nucleus), CD45 (for leukocytes) and ly6G (for neutrophils) to exclude the non-specific signals. These non-specific signals were observed from your single antibody staining. Beyond that, a basic H&E staining is necessitated to define the joint structure. H&E and multiple fluorescent IHC staining protocols are applied to all other imaging analysis in this manuscript.

Answer: For CD45 staining as suggested by the reviewer, this means that we will have to re-perform all of the immune staining, which will involve a great amount of work. Regarding the limited time frame for the revision, we could not finish doing it at this moment. But instead, we have provided

new images combining DAPI with Ly6G or Iba-1 immunostaining to show the co-localization of cell nucleus with these specific neutrophil or macrophage markers (Fig. 9J, Fig. 7G, Suppl. Fig. 3A&D, Suppl. Fig. 7C&E, Suppl. Fig. 10A). Some of the immunostaining quality have been improved in this revision as well. Ly6G and Iba-1 are both well-established markers for neutrophil and macrophage identification by immunostaining. Utilization of these markers alone (or combined with DAPI) in immunostaining is a common and simple method to identify inflammatory cell infiltration (e.g. some high profile journal papers, including PMID: 30804471, 31211699, 36754246, 32229220, 33608316 and more). In my lab, we also used the same Ly6G-EGFP knock-in mouse line as used in this study to successfully monitor neutrophil infiltration in ankle joints of gout arthritis model mice (Wei et al., Chin Med, 2023, PMID: 37464384). Here in this study, to avoid unspecific stained tissues, a cell is judged as Ly6G or Iba-1 positively labeled cell by simultaneously bearing 2 features: (1) stained positive for Ly6G or Iba-1; (2) having a definite DAPI-labeled nucleus. This method has been adopted in our recent study for the identification and calculation of Iba-1 positive microglia in spinal cord (Wang et al., J Neuroinflammation, 2023, PMID: 37158939). Using this method as above, we were able to obtain a relatively accurate calculation of infiltrated Ly6G and Iba-1 positive cells in periarticular tissues (bone area delineated by dashed line was excluded from calculation). The detailed analysis method has been added in Method section of this revised manuscript.

To further support the results obtained from immunostaining, in this revision, we continued to perform MPO activity tests in ankle joint or hind paw tissues. MPO test can quantify the amount of infiltrated neutrophils as MPO is a heme-containing peroxidase expressed mainly in neutrophils. We successfully used MPO assay in one of our recent studies to quantify neutrophil infiltrations in ankle joints of gout model mice (Yin et al., Br J Pharmacol, 2020, PMID: 31883118). The new MPO assay data have been added as Fig. 9C&L, Fig. 7F&I, and Suppl. Fig. 3C. Overall, the MPO results are all consistent with immunostaining or flow cytometry, which provides quantitative and parallel support to our major findings. We hope these efforts in all can address the reviewer's suggestions.

As suggested by the reviewer, we also provided H&E staining to further support our findings in Fig. 9. As seen, H&E staining offered an overview of the joint structure as well as periarticular tissues. It showed that gout arthritis model mice developed obvious signs of inflammatory cell infiltration in periarticular tissues vs. control mice. In contrast, *Cxcr2* cKO mice showed attenuated inflammatory cell infiltration (Suppl. Fig. 9). We further quantified the histopathological score of the ankle joint according to one recent study from Dr. Verri's group (PMID: 30914954). The score was determined by the sum of the 3 parameters: inflammatory infiltrate, cartilage injury and vascular proliferation. *Cxcr2^{fl/fl}*+MSU group mice showed increased histopathological score than control group, whereas SNS-Cre *Cxcr2^{fl/fl}*+MSU group mice showed significantly less score.

H&E staining further helped to demonstrate the important contribution of neuronal CXCR2 to immune modulation in ankle joints during gout arthritis. But as the reviewer knows, performing H&E staining of ankle joints is not an easy task, and usually takes a lot of steps, including decalcification, embedding, de-paraffinization, analysis, etc., which usually takes around two months or so to finish the whole process. If we have to provide H&E staining for all other experiments, this would require redoing virtually all of the experiments and we need to reestablish

the model, breed more knockout mice and re-collect ankle joint samples. More animals would thus have to be sacrificed for ankle joint collection that would be extremely time and resources consuming. Fortunately, we happened to preserve the ankle joint samples of control group and WT/*Cxcr2* cKO group mice in gout condition from our prior experiments. So we processed these tissues further for H&E staining as shown in new Suppl. Fig. 9. This part is also one of the key parts of this study showing *Cxcr2* cKO mice display reduced ankle inflammation vs. WT mice in gout arthritis condition. The H&E staining further helps to support this important conclusion. Although we cannot provide H&E staining for all other imaging at this moment, immunostaining has been used alternatively to delineate inflammatory cell infiltrations in ankle joints and hind paw tissues. Besides, we have incorporated multiple methods, including flow cytometry, qPCR, ELISA, MPO test and vascular permeation assay, etc. to investigate the detailed inflammatory condition of ankle joint. These methods in together provided solid supports to our major conclusions.

Again, we thank the reviewers for all these insightful comments that have helped to enhance the quality of this study. We hope our answers can address these suggestions. If other places in the manuscript still need to be improved, we will try our best to improve them. Thank you.

Reviewer #3 Comments:

The manuscript by Liu and co-authors showing us a very interesting and important work. CXCL5/neuronal CXCR2/TRPA1 axis contributes to gout arthritis pain, neutrophil influx and joint inflammation. Here, the study corroborates the CXCL5 overexpressed in animal models and serum from gout arthritis patients. Neuronal CXCR2 deletion was shown to ameliorate joint pain and inflammation in the animal model of gout arthritis. Moreover, neuronal CXCR2 or TRPA1 deletion prevented the hyperexcitability of DRG nociceptive sensory neurons induced by the CXCL5 administration. This study further revealed the neuronal CXCR2 coordinates with neutrophil CXCR2 to contribute to CXCL5-induced neutrophil chemotaxis by promoting CGRP and SP release and plasma extravasation in animals with gout arthritis. Additionally, this work first confirmed signaling of G $\beta\gamma$'s (CXCR2 signaling) contribution to activate TRPA1 by showing that G $\beta\gamma$ sequestration or G β knockdown reduced CXCR2-mediated TRPA1 activation.

Overall, Liu's work provide valuable evidence that CXCL5/CXCR2/TRPA1 signaling as a major player involved in gout arthritis pain. It will help us to further expand the knowledge of nociceptive sensory neurons-immune interactions in pain and inflammation. The manuscript is clearly written and easily readable. In my opinion the revised manuscript has well addressed the reviewer #2's comments. However, I have several minor concerns about the manuscript.

1. Fig. 5, the analyses of the electrophysiology data are solid but lack detail. More electrophysiological parameters for the DRG neurons (e.g. RMP, Cm, rheobase current...) should be presented in a table.

Answer: We first thank the reviewer for these positive comments on our manuscript and we all feel very encouraged. In this revision, we provided one table (Suppl. Table 2) to summarize some major electrophysiological parameters of DRG neurons under vehicle- and CXCL5-treated condition. However, we cannot provide rheobase current as mentioned, since the APs were elicited by step current as illustrated in Fig. 5A. We did not use the slope current for stimulation, thus it is impossible

to deduce the rheobase current.

2. Line 742, AP were evoked by a 600-ms step current, while in the Fig 5C, the schematic diagram marked as 500-ms of step current.

Answer: Sorry for this inconstancy. We rechecked our stimulating protocol and confirmed that the AP was induced by 500-ms step current. This inconsistency has been modified in the manuscript.

3. Line 773-774, is there any evidence or literature references to support the $C_m < 42$ pF is a small diameter DRG neuron?

Answer: In a previous study, people have systematically checked the relationship of size distribution with membrane capacitance and found that almost all of the small-sized DRG neurons showed $C_m < 44$ pF (PMID: 25431155). In another study, small-sized DRG neuron was chosen based on $C_m < 42$ pF (Lopez-Santiago et al., J Neurosci. 2006, PMID: 16870743). Thus, we adopted $C_m < 42$ pF as the criteria for selecting small-sized DRG neuron in this study. In addition, our lab have successfully adopted the same criteria in our recent studies (Hu et al., Front Pharmacol, 2019, PMID: 31105572 & Wang et al., J Adv Res, 2021, PMID: 34194835) and found that DRG neurons with $C_m < 42$ pF showed very high responsive rate to capsaicin (nearly 70% in control condition and over 80% in chronic pain condition) and showed obvious Nav1.7 & Nav1.8 expression, further supporting the practicability of this criteria. The relevant supporting literatures have been added in methods section of this revised manuscript. Thanks.

4. The present results shown that the CXCL1 or CXCL2 mediate inflammation (Fig1I&J) via the receptor CXCR2, and CXCL1 also produced Ca^{2+} response in mouse DRG neurons (Fig 3F). These points should be addressed in the discussion.

Answer: Thanks for the suggestion. We found that neutralizing CXCL1 or CXCL2 only showed ameliorating effects on ankle joint inflammation, but not on joint pain, whereas neutralizing CXCL5 ameliorates both pain and ankle inflammation of gout arthritis model mice. CXCL1, 2 and 5 bind with CXCR2 to produce biological effects. We used Ca^{2+} imaging to further compare their abilities to excite DRG neurons. We found that among these chemokines, CXCL5 produced the most obvious DRG neuron activation, which was significantly stronger than CXCL1. In contrast, CXCL2 produced no activation at all. We further confirmed that both CXCL5 and CXCL1's activating effects on DRG neurons were CXCR2-dependent. This finding implies that CXCL5 can be more powerful than CXCL1 or CXCL2 in producing nociceptive signals during gout arthritis. RNA-Seq further reveals that CXCL5 expression is the most upregulated among these chemokines in ankle joints of model mice. Regarding the relatively high expression of CXCL5 in the inflamed ankle joints and its high potency in exciting sensory neurons, neutralization of CXCL5 can produce the most effective ameliorating effect on both pain and inflammation in gout model mice. Further studies will be needed to thoroughly investigate the mechanisms underlying different potencies of these chemokines to activate neuronal CXCR2. This part has been added to the discussion of this revised manuscript.

REVIEWER COMMENTS

Reviewer #1 (Remarks to the Author):

Thank you for the revised version with elaboration that the key mechanism of neutrophilic influx is via neuronal CXCR2 as demonstrated by the selective neuronal CXCR2 KO mice experiments, whilst other contributory mechanism eg via neutrophilic CXCR2 axis remains intact. This raises the following question: when 300ng CXCL5 was administered via intra-articular injection, why was neutrophil joint infiltration reduced in CXCR2 selectively knockout mice (SNS-Cre CXCR2fl/fl) , given that CXCL5 blood-tissue chemokine gradient dictates neutrophil trafficking. One would expect that neutrophil infiltration to ankle joint would be similar between CXCR2 selectively knockout mice (SNS-Cre CXCR2fl/fl) and control mice (CXCR2fl/fl) as the CXCL5 blood-tissue chemokine gradient is similar. To further understand the underlying mechanism, you might consider

1) measuring the serum levels of CXCL5 before and after high dose CXCL5 intra-articular injection in both SNS-Cre CXCR2fl/fl and CXCR2fl/fl mice

2) dynamic joint neutrophil count within 24 hours, such as at 0h, 2h, 6h, 16h and 24h. this will help to address whether the neutrophil infiltration is similar in both mice models in the first few hours and start to reduce at a later phase or whether the reduced neutrophil infiltration occurs at the beginning. The time of measurement of neutrophil joint infiltration was not stated.

The above would help in our understanding of whether and how neuronal CXCR2 KO alters the blood tissue CXCL5 gradient and neutrophil trafficking

The reply to the 2nd comment has been reasonably addressed

Reviewer #3 (Remarks to the Author):

The author has answered all of my questions. I have no further concern.

Reviewer #1 (Remarks to the Author):

Thank you for the revised version with elaboration that the key mechanism of neutrophilic influx is via neuronal CXCR2 as demonstrated by the selective neuronal *Cxcr2* KO mice experiments, whilst other contributory mechanism eg via neutrophilic CXCR2 axis remains intact. This raises the following question: when 300ng CXCL5 was administered via intra-articular injection, why was neutrophil joint infiltration reduced in CXCR2 selectively knockout mice (SNS-Cre *Cxcr2*^{fl/fl}), given that CXCL5 blood-tissue chemokine gradient dictates neutrophil trafficking. One would expect that neutrophil infiltration to ankle joint would be similar between CXCR2 selectively knockout mice (SNS-Cre *Cxcr2*^{fl/fl}) and control mice (*Cxcr2*^{fl/fl}) as the CXCL5 blood-tissue chemokine gradient is similar. To further understand the underlying mechanism, you might consider 1) measuring the serum levels of CXCL5 before and after high dose CXCL5 intra-articular injection in both SNS-Cre *Cxcr2*^{fl/fl} and *Cxcr2*^{fl/fl} mice.

2) dynamic joint neutrophil count within 24 hours, such as at 0h, 2h, 6h, 16h and 24h. this will help to address whether the neutrophil infiltration is similar in both mice models in the first few hours and start to reduce at a later phase or whether the reduced neutrophil infiltration occurs at the beginning. The time of measurement of neutrophil joint infiltration was not stated.

The above would help in our understanding of whether and how neuronal *Cxcr2* KO alters the blood tissue CXCL5 gradient and neutrophil trafficking

The reply to the 2nd comment has been reasonably addressed

For questions, please see our response as below:

1) Measuring the serum levels of CXCL5 before and after high dose CXCL5 intra-articular injection in both SNS-Cre CXCR2^{fl/fl} and CXCR2^{fl/fl} mice.

Answer: We thank the reviewer for the positive comments for our manuscript. In our study, we found that CXCL5 injection can produce robust neutrophil infiltration (Fig. 7F-H), as revealed by MPO and immunostaining. Interestingly, CXCL5-induced neutrophil infiltration was significantly reduced in SNS-Cre *Cxcr2*^{fl/fl} mice. To further understand the underlying mechanisms, we showed that CXCL5 could activate CXCR2 expressed in nociceptors (Fig. 3&5) and induced CGRP and SP release (Fig. 7D&E, K&L), two potent neuropeptides involved in vasodilation and plasma extravasation. CXCL5-induced CGRP and SP release further promoted plasma extravasation, as evidenced by the reduction in Evans blue extravasation in hind paw upon CXCL5 injection in SNS-Cre *Cxcr2*^{fl/fl} mice or in mice treated with CGRP or SP receptor antagonist (Fig. 7A-C). These results in all suggest that neuronal CXCR2 contributes to CXCL5-induced neutrophil infiltration by promoting CGRP and SP release. These two neuropeptides in turn cause vasodilation and plasma extravasation that facilitate CXCL5/neutrophilic CXCR2-induced neutrophil infiltration.

The reviewer suggests us to measure serum levels of CXCL5 before and after high dose CXCL5 injection in both SNS-Cre *Cxcr2*^{fl/fl} and *Cxcr2*^{fl/fl} mice. Accordingly, we measured the serum levels of CXCL5 before and 3 h after CXCL5 injection (300 ng) in both SNS-Cre *Cxcr2*^{fl/fl} and *Cxcr2*^{fl/fl} mice. ELISA showed that 300 ng CXCL5 local injection did not significantly affect serum CXCL5

level in either *Cxcr2^{fl/fl}* or SNS-Cre *Cxcr2^{fl/fl}* (see new Suppl. Fig. 7G). This means that locally injected 300 ng CXCL5 exerts its effect mainly in local tissues but did not produce significant systematic CXCL5 upregulation in both group of mice. Therefore, this experiment rules out the possibility that neuronal CXCR2 cKO may upregulate blood CXCL5 level to compromise neutrophil infiltration, which further supports our original conclusion.

2) Dynamic joint neutrophil count within 24 hours, such as at 0h, 2h, 6h, 16h and 24h. this will help to address whether the neutrophil infiltration is similar in both mice models in the first few hours and start to reduce at a later phase or whether the reduced neutrophil infiltration occurs at the beginning. The time of measurement of neutrophil joint infiltration was not stated.

Answer: We are sorry that we did not clearly state the exact time points for neutrophil measurement in our original manuscript. Actually, we collected the hind paw tissues for assay 3 h after CXCL5 or vehicle injection as shown in Fig. 7A-H. This info has been added in figure legend. In this revision, in order to further measure the dynamic changes in neutrophil infiltration by CXCL5 as suggested, we collected hind paw tissues of both SNS-Cre *Cxcr2^{fl/fl}* and *Cxcr2^{fl/fl}* mice at 0, 3, 6, 16 and 24 h time points and performed MPO assay. The new results showed that CXCL5 injection caused obvious neutrophil infiltration at 3 and 6 h time points in *Cxcr2^{fl/fl}* mice and the response near completely diminished at 16 h and later (24 h) time points. More importantly, CXCL5-induced neutrophil infiltration was significantly reduced in SNS-Cre *Cxcr2^{fl/fl}* at both 3 and 6 h time points (see new Fig. 7F). This result indicates that the deficit in neutrophil infiltration observed in SNS-Cre *Cxcr2^{fl/fl}* mice occurs both at the beginning (3 h) and at a later phase (6 h) after CXCL5 injection. This dynamic neutrophil measurement in both SNS-Cre *Cxcr2^{fl/fl}* and *Cxcr2^{fl/fl}* mice further improves our study and provides solid support to our conclusions. We thank the reviewer for all these insightful comments and we hope our answers can satisfactorily address these questions.

REVIEWERS' COMMENTS

Reviewer #1 (Remarks to the Author):

Thank you for the further experiments, which have adequately addressed the queries and clarifications, and strengthens the study